# Removal of Metal Ions via Adsorption Using Carbon Magnetic Nanocomposites: Optimization through Response Surface Methodology, Kinetic and Thermodynamic Studies

Simona Gabriela Muntean [1,*] , Liliana Halip [1,*], Maria Andreea Nistor [1] and Cornelia Păcurariu [2]

[1] "Coriolan Drăgulescu" Institute of Chemistry Timisoara of Romanian Academy, 300223 Timisoara, Romania; nistor_andreea1990@yahoo.com

[2] Faculty of Industrial Chemistry and Environmental Engineering, Politehnica University Timisoara, P-ța Victoriei No. 2, 300006 Timisoara, Romania; cornelia.pacurariu@upt.ro

* Correspondence: sgmuntean@acad-icht.tm.edu.ro (S.G.M.); lili.ostopovici@gmail.com (L.H.)

**Abstract:** The toxicity of metal ions on ecosystems has led to increasing amounts of research on their removal from wastewater. This paper presents the efficient application of a carbon magnetic nanocomposite as an adsorbent for the elimination of metal ions (copper, lead and zinc) from aqueous solutions. A Box–Behnken factorial design combined with the response surface methodology was conducted to investigate the effect and interactions of three variables on the pollutant removal process. Highly significant ($p < 0.001$) polynomial models were developed for each metal ion: the correlation coefficient was 0.99 for Cu(II) and Pb(II), and 0.96 for Zn(II) ion removal. The experimental data were in agreement and close to the theoretical results, which supports the applicability of the method. Working at the natural pH of the solutions, with a quantity of carbon magnetic nanocomposite of 1 g/L and a metal ions' concentration of 10 mg/L, for 240 min, removal efficiencies greater than 75% were obtained. The kinetic study indicated that a combination of kinetic models pseudo-second order and intraparticle diffusion were applied appropriately for copper, lead and zinc ion adsorption on carbon magnetic nanocomposite. The maximum adsorption capacities determined from the Langmuir isotherm model were 81.36, 83.54 and 57.11 mg/g for copper, lead and zinc ions. The average removal efficiency for five adsorption–desorption cycles was 82.21% for Cu(II), 84.50% for Pb(II) and 72.68% for Zn(II). The high adsorption capacities of metal ions, in a short time, as well as the easy separation of the nanocomposite from the solution, support the applicability of the magnetic carbon nanocomposite for wastewater treatment.

**Keywords:** response surface methodology; adsorption; metal ions; kinetic; isotherm

## 1. Introduction

Contamination of water with various pollutants, such as metal ions, has become a major environmental and health problem many countries face, which poses a threat to society and living organisms [1–3]. Metal ions are reported to be pollutants due to their mobility in natural water ecosystems and due to their toxicity [4]. According to the WHO, the permissible limits for the presence of heavy metals investigated in wastewater are as follows: 0.02 mg/L for copper, 0.01 for lead, and 3.00 for zinc [5]. The problem associated with metal ions pollution is that they are not biodegradable, cannot be metabolized or decomposed into compounds with reduced toxicity, and are highly persistent in the environment [6,7]. In this context, the availability of clean water used for various activities has become a challenging task for researchers [2,8,9].

Wastewater produced by many industries (textiles, paper, food, cosmetics) must be treated before being discharged into the environment or sewage. The treatment of this wastewater must be carried out using robust, simple, economically feasible and environmentally friendly techniques [4,10,11]. The most widely used techniques for improving

effluent quality include the following: membrane filtration, coagulation and flocculation, chemical precipitation, adsorption, ion exchange, electrochemical removal, reverse osmosis and oxidation processes [11–13]. Most of these methods involve high operational and maintenance costs. Adsorption has proven to be a cost-effective and efficient technique for removing dyes and metal ions from wastewater [4,14–17].

After extensive use in wastewater treatment, activated carbon has proven to be an effective absorbent material, but it presents many disadvantages, including the following: high production costs, low mechanical properties, and, most importantly, its difficult separation from the solution, which causes problems of regeneration and reuse [18].

In this work, a new nanocomposite material based on magnetic iron oxide, silver and activated carbon was synthesized using an innovative combustion technique [19]. The specific surface area of the nanocomposite (NC) is very close to that of the activated carbon but the major advantage is given by its magnetic characteristics [16,20–23]. Therefore, using this NC, the phase separation occurs more easily, while the adsorption properties remain excellent. Additionally, the possibility to regenerate and reuse the NC in other experiments must not be neglected [24,25].

In the event of using the NC on industrial scale, we proceed to identify the optimal conditions for the most efficient removal of three metal ions, as important pollutants, from wastewater: Cu(II), Pb(II) and Zn(II). The adsorption process is controlled by several factors depending on the pollutant type, wastewater source, etc. The number of experiments that need to be performed in order to identify the optimal conditions is highly dependent on the number of parameters that directly affect the adsorption process. Thus, the higher the number of process parameters, the more experiments are required as the response must be measured using all the possible combinations of the factors that influence the adsorption. Factorial design methods represent a reasonable alternative to save time as well financial and material resources. In the present work, we combined a Box–Behnken design and response surface methodology to optimize the removal of Cu(II), Pb(II) and Zn(II) from wastewater [26,27]. These methods have been shown to be successful when applied to predict the optimal working conditions of adsorption studies/processes [28,29]. The selected process parameters were the solution's pH, initial metals ions' concentration and NC dosage. For each pollutant, we performed a limited number of experiments, which were used further to build mathematical models that best describe the removal efficiency. The parameters with a significant influence on the adsorption were identified, together with the interactions between them. Finally, the interaction between the factors was studied and optimized using response surface methodology (RSM). The predicted conditions for an efficient removal were experimentally validated. The adsorption capacities of the adsorbent for the removal of metal ions from solutions were determined via isotherm studies. Additionally, kinetics and thermodynamic studies on the adsorption process were performed.

## 2. Experimental

### 2.1. Materials and Method

The following chemicals of analytical grade were used without further purification: iron nitrate nonahydrate ($Fe(NO_3)_3 \cdot 9H_2O$, Carl Roth, Karlsruhe, Germany), granular activated carbon (Oltchim, Râmnicu Vâlcea, Romania), silver nitrate ($AgNO_3$, Merck, Darmstadt, Germany), and citric acid monohydrate ($C_6H_8O_7 \cdot H_2O$, Merck, Darmstadt, Germany).

The NC magnetic nanocomposite was synthesized using the combustion method, described in detailed in our previous work [19], using 25.40 g iron nitrate and 3.28 g of silver nitrate as oxidizing agents, 12.60 g citric acid as reducing agent, and 34.73 carbon. Briefly, the flask containing the mixture of activated carbon impregnated with the solution containing iron nitrate, silver and citric acid was placed inside a heating mantle at 400 °C and maintained for 60 min. The smoldering combustion reaction was accompanied by the release of a large amount of gases that were bubbled into a water vessel, with the reaction taking place in a controlled atmosphere (in the absence of air). The resulting powders were

washed with distilled water and dried at 60 °C. The recipe was calculated for a mass ratio iron oxide (calculated as $Fe_3O_4$)–silver–carbon of 0.7:0.3:5.

Copper chloride ($CuCl_2$), lead chloride ($PbCl_2$), and zinc chloride ($ZnCl_2$) were purchased from Merck.

The stock solutions of the metal ions used in the adsorption studies were prepared by dissolving an appropriate amount of salt (weighed on the analytical balance) in distilled water and diluting it in a volumetric flask. The working solutions were prepared by diluting with distilled water an exact volume of the stock solutions so that the concentration of the metal ion in the working solution was within the concentration range studied, and the pH was adjusted using HCl or NaOH solutions (0.1 N).

### 2.2. Characterization of NC

The phase composition of the sample was via by X-ray diffraction (XRD) using a Rigaku Ultima IV diffractometer equipped with CuKα radiation (λ = 1.54059 Å) in the 2θ range 15°–80°, at 40 kV, 40 mA. Thermogravimetry coupled with differential scanning calorimetry (TG-DSC) was performed with a Netzsch STA 449C instrument under dynamic air atmosphere at a flow rate of 200 mL $min^{-1}$ with a heating rate of 10 K $min^{-1}$ using Pt crucibles. The morphology of the nanocomposite was investigated via scanning electron microscopy (SEM) and energy dispersive X-ray (EDX) using a FEI Quanta FEG 250 microscope. $N_2$ adsorption–desorption isotherms were performed using a Micromeritics ASAP 2020 instrument at liquid nitrogen temperature. The powder was previously degassed under high vacuum (5 μm Hg) for 8 h at 100 °C. The specific surface area of the sample, $S_{BET}$, was measured using the BET (Brunauer, Emmett, and Teller) method and the pore size distribution was computed using the Barrett–Joyner–Halenda (BJH) method from the desorption curves. The behavior in the external magnetic field of the sample was studied at room temperature under AC (50 Hz) applied magnetic fields up to 300 kA $m^{-1}$ using a laboratory installation equipped with a data acquisition system.

### 2.3. Adsorption and Desorption Experiments

The efficiency of the synthesized magnetic nanocomposite (NC) as an adsorbent was tested in batch experiments. The NC was added to the metal ions solution, stirred at 200 rpm, at different working conditions, until equilibrium was reached. After magnetic separation of the adsorbent from the solution, the residual concentration of metal ions in the solution was determined using atomic absorption spectrophotometry (SensAA).

The adsorption capacity at equilibrium ($q_e$), and the metal ion removal efficiency ($R$) were calculated using concentrations determined from the Lambert–Beer Law:

$$q_e(\text{mg/g}) = \frac{\left(initial\ conc.\ of\ metal\ ion\left(\frac{\text{mg}}{\text{L}}\right) - metal\ ion\ conc.\ at\ equilibrium(\text{mg/L})\right) \cdot volume(\text{L})}{NC\,mass(g)} \qquad (1)$$

$$R = \frac{\left(initial\ conc.\ of\ metal\ ion\left(\frac{\text{mg}}{\text{L}}\right) - conc.\ of\ metal\ ion\ at\ equilibrium(\text{mg/L})\right) \cdot 100}{initial\ conc.\ of\ metal\ ion(\text{mg/L})} \qquad (2)$$

Desorption studies were performed in order to evaluate the reusability of NC as an adsorbent, using 50 mL HCl solution (0.5 N).

### 2.4. Factorial Design

A three-level Box–Behnken factorial design with three factors (process parameters) was conducted for each pollutant. The solution pH, the initial metal ion concentration ($C_{Cu}$; $C_{Pb}$; $C_{Zn}$) and the NC dosage ($D_{NC}$) were studied as the main factors influencing the metal ions' removal process. For each pollutant, the variation interval of the factors was defined by setting the upper, central and lower limits, which were subsequently coded with the levels +1, 0 and −1, respectively. The number of performed experiments, N, is as shown in Equation (3), where k represents the number of factors and $n_0$ the number of replicated experiments at the central point. The central point estimates the medians of the values used

in the factorial range (coded level 0) and it is replicated in order to detect the precision of the experiments.

$$N = 2 \cdot k \cdot (k - 1) + n_0 \tag{3}$$

According to Equation (3), we run a number of $2 \times 3 \times 2 + 3 = 15$ experiments. The selected variables and their limits, units and notations are shown in Table 1.

**Table 1.** The setting for the process variables and their limits used in full-factorial experiment.

| Factor | Variable | Unit | Level | | |
|---|---|---|---|---|---|
| | | | **−1** | **0** | **+1** |
| X1 | pH | - | 4 | 6 | 8 |
| X2 | Initial metal concentration ($C_{cu}$; $C_{Pb}$; $C_{Zn}$) | mg/L | 10 | 105 | 200 |
| X3 | NC dosage ($D_{NC}$) | g/L | 0.5 | 1 | 1.5 |

### 2.5. Statistical Analysis

The experimental design and the statistical analysis were conducted using the *rsm* package in R [30]. The relationship between the response variable and the factors could be described using a quadratic model as in Equation (4), where $Y$ represents the output (yield), $i$ and $j$ are index numbers for factors, $\beta_0$ is the free term (intercept), $\beta_i$ represents the regression coefficients for linear effects, $\beta_{ii}$ represents the regression coefficients for the quadratic effect, $\beta_{ij}$ represents the regression coefficients for the interaction effects, $X_i$ represents the coded values for the experimental variables, and $\varepsilon$ represents the random error.

$$Y = \beta_0 + \sum \beta_i X_i + \sum \beta_{ij} X_i X_j + \sum \beta_{ij} X_i^2 + \varepsilon \tag{4}$$

The significance of the regression model, the effect of each factor and their interactions were evaluated with the analysis of variance (ANOVA). The tests were considered significant if the *p*-value was less than 0.05. The contour plots and response surfaces were generated using the fitted quadratic polynomial equation.

## 3. Results and Discussions

### 3.1. Characterization of NC Adsorbent

The X-ray diffraction pattern of the NC nanocomposite illustrated in Figure 1 reveals the lack of the activated carbon diffraction peaks, which confirms its amorphous structure in the sample.

The following COD files were used for the peaks' assignment: COD-9002318 (magnetite, $Fe_3O_4$) COD-9012692 (maghemite, $\gamma$-$Fe_2O_3$), and COD-9011607 (silver, Ag). Because both magnetite and maghemite crystallize in the cubic system and their peaks are located at very close values of $2\theta$, it is quite difficult to differentiate between the two phases. The most applied method to differentiate between magnetite and maghemite is X-ray diffraction; however, more accurate techniques such as the Mössbauer [31] and X-ray photoelectron spectroscopy [32] have also begun being applied.

Based on the X-ray diffraction pattern (Figure 1), the mass percentage of crystalline phases in the sample was estimated with PDXL2 software (version PDF 4+) using the relative intensity ratio (RIR) method. It was established that magnetite (68%) is the main phase in the magnetic nanocomposite, alongside maghemite (7%) and silver (25%).

It is also worth noting the sharp peaks exhibited by the X-ray diffraction pattern (Figure 1), which confirms the high crystallinity of $Fe_3O_4$/$\gamma$-$Fe_2O_3$ and Ag in the NC nanocomposite.

The TG-DSC curves of the magnetic nanocomposite are shown in Figure 2.

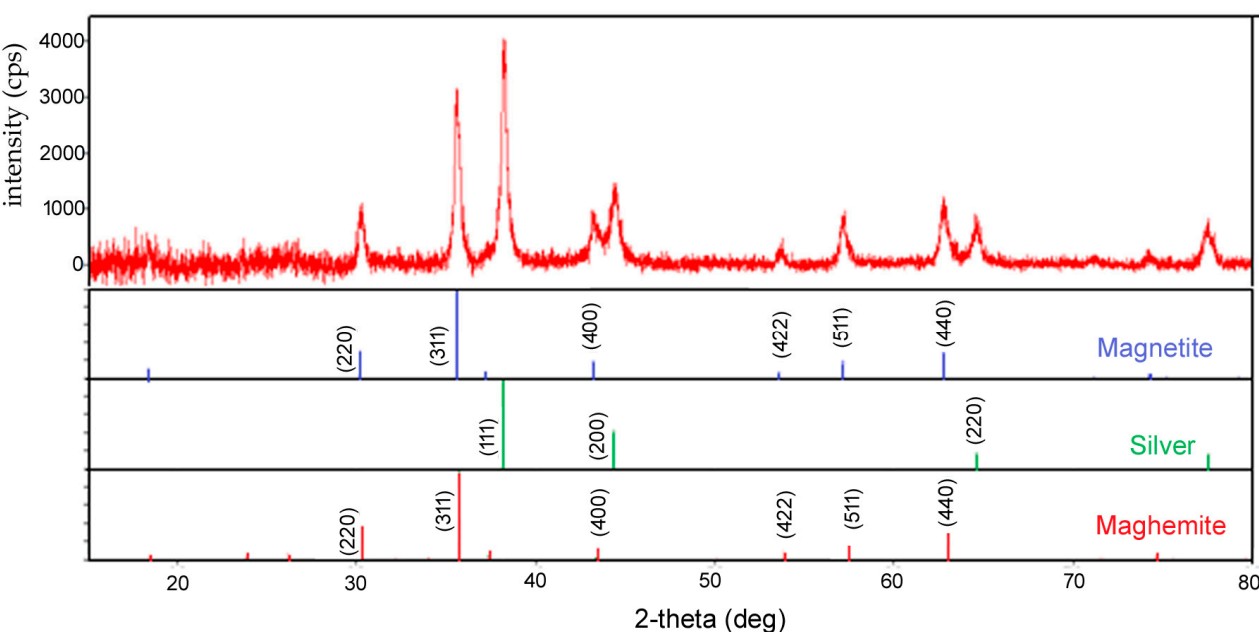

**Figure 1.** XRD pattern of the NC nanocomposite.

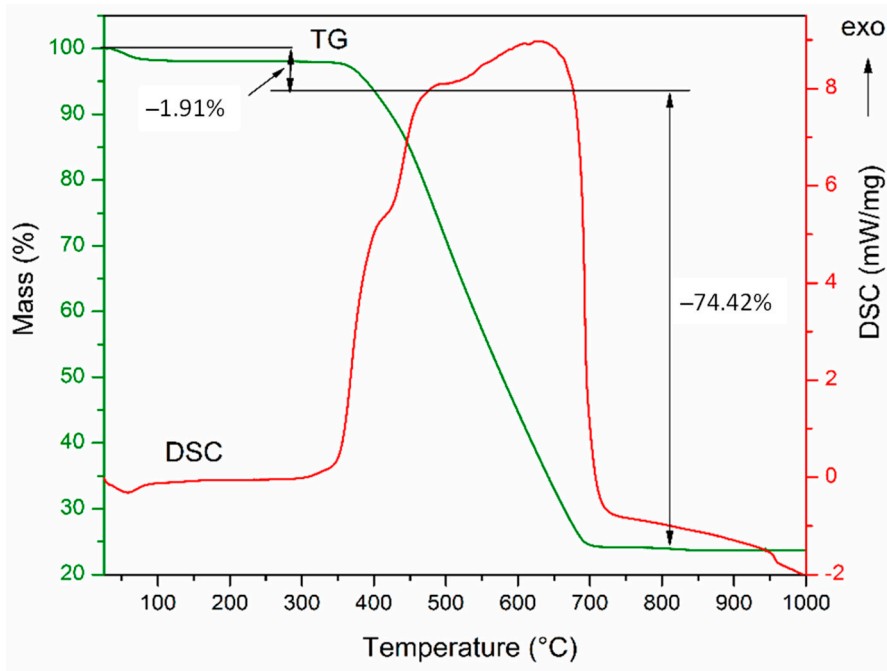

**Figure 2.** TG-DSC curves of NC sample.

Alongside a small endothermic effect on DSC curve at about 60 °C, accompanied by 1.91% mass loss, which corresponds to the water evaporation, a large exothermic effect between 300 and 750 °C can be observed, accompanied by a significant mass loss on the TG curve (74.42%). The strong exothermic effect overlaps multiples exothermic effects attributed to the burning of organics residuals resulted from the citric acid used as fuel, as well as that of the carbon present in the sample.

The SEM analysis of the nanocomposite shown in Figure 3 reveals small particles of magnetic iron oxide and silver of irregular shape that are agglomerated as aggregates on the large surface of carbon.

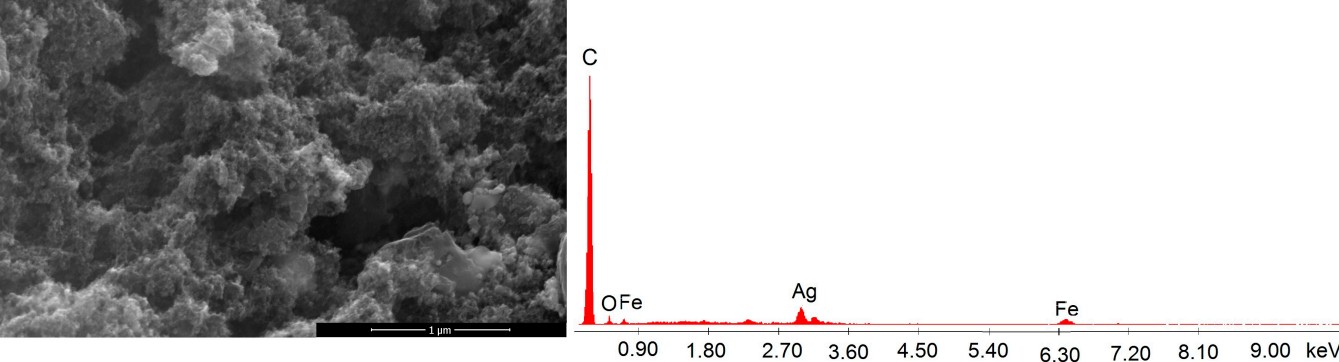

**Figure 3.** SEM-EDX analysis of NC sample.

The EDX spectrum evidences the high content of carbon alongside iron, oxygen and silver, in accordance with the sample composition (the unmarked peaks correspond to aluminum—sample holder material).

The $N_2$ adsorption–desorption isotherms of the NC nanocomposite presented in Figure 4a exhibits a type II profile with H3 hysteresis according to the IUPAC classification [33].

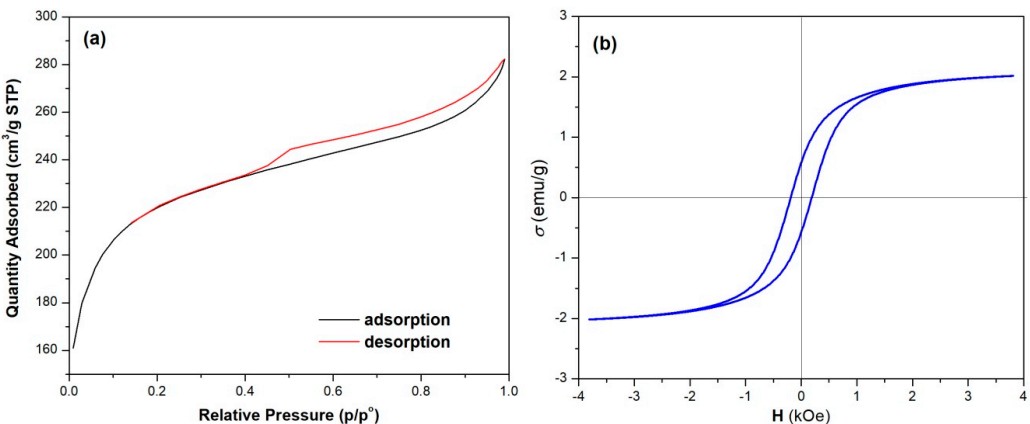

**Figure 4.** (**a**) $N_2$ adsorption–desorption isotherms of NC sample; (**b**) magnetization curves of NC sample.

In order to estimate the adsorption capacity of the magnetic NC nanocomposite, the following textural characteristics of the sample were determined: BET specific surface area 755.2 $m^2/g$, desorption cumulative pore volume 0.181 $cm^3/g$, desorption average pore diameter 4.05 nm, and desorption cumulative surface area of pores 178.67 $m^2/g$.

The specific surface area of NC (755.2 $m^2/g$) being very close to that of carbon (889.9 $m^2/g$) indicates that the magnetic nanomaterial has very good adsorption capacity, which indicates it may be used as an adsorbent for the removal of different pollutants from wastewaters.

The magnetization curve of the synthetized nanocomposite is presented in Figure 4b.

The low value of the saturation magnetization of 2.0 emu/g is due to the large amount of non-magnetic carbon present in the NC nanocomposite. The hysteresis loop of the sample can be observed, which evidences a remanent magnetization of 0.5 emu/g, which indicates that the nanocomposite does not have superparamagnetic properties. Despite the low saturation magnetization value, the NC nanocomposite can be easily separated from aqueous solutions with a magnet.

### 3.2. Factorial Design

A Box–Behnken design (BBD) was carried out to establish the influence of experimental variables (factors) and their interactions on the metal ions' removal. The BBD comprises

twelve factorial and three central points; therefore, a total of 15 runs must be used to determine the optimum experimental conditions. The design matrix of the experiments and the experimental results for Cu(II), Pb(II), and Zn(II) removal efficiency are presented in Table 2. The standard deviations of the average yields for the central points are 0.59% for Cu(II) removal, 0.50% for Pb(II) removal, and 1.12% for Zn(II) removal.

**Table 2.** The experimental results for Cu(II), Pb(II), Zn(II) removal using NC.

| Run | $X_1$ * | $X_2$ * | $X_3$ * | $Y_{Cu}$ (%) | $Y_{Pb}$ (%) | $Y_{Zn}$ (%) |
|---|---|---|---|---|---|---|
| 1 | −1 | −1 | 0 | 49.30 | 56.32 | 34.90 |
| 2 | +1 | −1 | 0 | 84.79 | 88.65 | 74.14 |
| 3 | −1 | +1 | 0 | 18.47 | 21.34 | 15.42 |
| 4 | +1 | +1 | 0 | 38.16 | 43.56 | 29.34 |
| 5 | −1 | 0 | −1 | 19.88 | 29.14 | 17.48 |
| 6 | +1 | 0 | −1 | 50.21 | 52.65 | 28.64 |
| 7 | −1 | 0 | +1 | 41.12 | 50.26 | 36.97 |
| 8 | +1 | 0 | +1 | 72.65 | 76.45 | 60.25 |
| 9 | 0 | −1 | −1 | 57.90 | 72.14 | 36.70 |
| 10 | 0 | +1 | −1 | 16.98 | 18.72 | 10.12 |
| 11 | 0 | −1 | +1 | 77.80 | 88.31 | 69.20 |
| 12 | 0 | +1 | +1 | 42.00 | 30.45 | 21.87 |
| 13 | 0 | 0 | 0 | 45.22 | 57.21 | 38.90 |
| 14 | 0 | 0 | 0 | 44.43 | 58.01 | 37.66 |
| 15 | 0 | 0 | 0 | 45.58 | 58.12 | 39.89 |

* as in Table 1; $Y_{Cu}$, $Y_{Pb}$, $Y_{Zn}$ represent the measured yields for Cu(II), Pb(II) and Zn(II) removal.

### 3.2.1. The Influence of the Main Effect

The experimental data provide a first clue regarding the influence of each factor on the pollutant removal process. The main effect of the working parameters on the removal efficiency of Cu(II) Zn(II) and Pb(II) is shown in Figure 5. All three parameters were found to significantly influence the efficiency of the removal of metal ions. The metal ions' removal is negatively influenced by the metal ions' concentration, ($C_{Cu}$, $C_{Pb}$, and $C_{Zn}$) as the efficiency decreases from the lowest to the highest level. In contrast, the pH and sorbent dose have a positive effect on the removal efficiency as they induce an efficiency increase from the lowest to the highest level.

### 3.2.2. Model Generation

An exhaustive regression analysis was performed in order to obtain the best empirical model for Cu(II), Pb(II), and Zn(II) removal efficiency based on the experimental design matrix presented in Table 2. The best fitted models that describe the relationships between the removal efficiency of metal ions and the independent variables are quadratic (copper and lead) and linear (zinc) polynomial models, which are as shown in coded Equations (5)–(7).

$$R_{Cu} = 46.61 + 2.59 \cdot X_2^2 \; 14.83 \cdot X_1 - 19.54 \cdot X_2 + 11.07 \cdot X_3 - 3.95 \cdot X_1 \cdot X_2 \quad (5)$$

$$\begin{aligned} R_{Pb} = 57.27 - 4.98 \cdot X_1^2 + 13.03 \cdot X_1 - 20.41 \cdot X_2 + 10.35 \cdot X_3 \\ -2.52 \cdot X_1 \cdot X_2 + 2.89 \cdot X_2 \cdot X_3 \end{aligned} \quad (6)$$

$$R_{Zn} = 36.76 + 10.95 \cdot X_1 - 17.27 \cdot X_2 + 11.92 \cdot X_3 - 6.33 \cdot X_1 \cdot X_2 - 5.18 \cdot X_2 \cdot X_3 \quad (7)$$

The corresponding statistical regression coefficients are presented in Table 3.

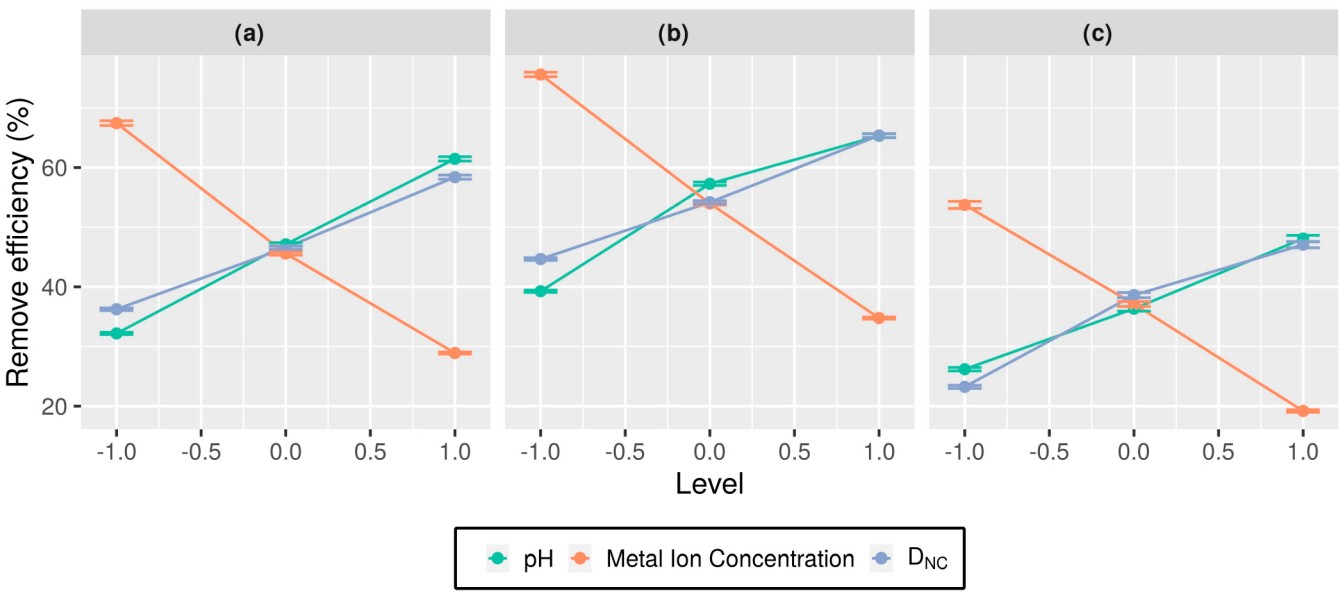

**Figure 5.** The effects of operational parameters against the removal efficiency of: (**a**) Cu(II), (**b**) Pb(II), and (**c**) Zn(II) ions.

**Table 3.** Statistical regression coefficients for Cu(II), Pb(II), and Zn(II) removal efficiency.

| Pollutant | | Coefficient | Std. Error | *T* Value | *p*-Value | Model Significance |
|---|---|---|---|---|---|---|
| Cu(II) | (Intercept) | 46.61 | 0.52 | 90.39 | $1.25 \times 10^{-14}$ | $R^2 = 0.9971$<br>$R^2$adj = 0.9955<br>F-value = 616.4<br>*p*-value = $3.99 \times 10^{-11}$ |
| | $(X_2)^2$ | 2.59 | 0.70 | 3.67 | $5.18 \times 10^{-3}$ | |
| | $X_1$ | 14.84 | 0.48 | 30.63 | $2.06 \times 10^{-10}$ | |
| | $X_2$ | −19.54 | 0.49 | −39.99 | $1.92 \times 10^{-11}$ | |
| | $X_3$ | 11.08 | 0.48 | 22.93 | $2.70 \times 10^{-9}$ | |
| | $X_1$:$X_2$ | −3.95 | 0.68 | −5.78 | $2.65 \times 10^{-4}$ | |
| Pb(II) | (Intercept) | 57.28 | 0.55 | 104.13 | $8.08 \times 10^{-14}$ | $R^2 = 0.997$<br>$R^2$adj = 0.994<br>F-value = 448.8<br>*p*-value = $1.15 \times 10^{-9}$ |
| | $(X_1)^2$ | −4.98 | 0.75 | −6.62 | $1.66 \times 10^{-4}$ | |
| | $X_1$ | 13.03 | 0.51 | 25.33 | $6.32 \times 10^{-9}$ | |
| | $X_2$ | −20.42 | 0.51 | −39.68 | $1.78 \times 10^{-10}$ | |
| | $X_3$ | 10.35 | 0.51 | 20.12 | $3.88 \times 10^{-8}$ | |
| | $X_1$:$X_2$ | −2.53 | 0.73 | −3.47 | $8.40 \times 10^{-3}$ | |
| | $X_2$:$X_3$ | 2.89 | 0.73 | 3.97 | $4.11 \times 10^{-3}$ | |
| Zn(II) | (Intercept) | 36.77 | 1.05 | 35.00 | $6.26 \times 10^{-11}$ | $R^2 = 0.9696$<br>$R^2$adj = 0.9527<br>F-value = 57.41<br>*p*-value = $1.48 \times 10^{-6}$ |
| | $X_1$ | 10.95 | 1.44 | 7.61 | $3.28 \times 10^{-5}$ | |
| | $X_2$ | −17.27 | 1.44 | −12.01 | $7.64 \times 10^{-7}$ | |
| | $X_3$ | 11.92 | 1.44 | 8.29 | $1.66 \times 10^{-5}$ | |
| | $X_1$:$X_2$ | −6.33 | 2.03 | −3.11 | $1.25 \times 10^{-2}$ | |
| | $X_2$:$X_3$ | −5.19 | 2.03 | −2.55 | $3.12 \times 10^{-2}$ | |

Encoded values can be easily converted to real values and vice versa using Equation (8).

$$X_i = \frac{(x_i - x_0)}{\Delta x} \tag{8}$$

where xi and xi are the coded and the real values of factor xi, respectively, $x_0$ is the actual value of the independent variable at the central point, and $\Delta x$ is the step change of xi corresponding to a unit variation of the dimensionless value.

From the analysis of Equations (5)–(7), it can be observed that the obtained models are in agreement with the observations resulting from the analysis of the experimental measurements: pH and $D_{NC}$ terms have a positive sign of the effect, which designates a greater influence of the variable at a higher level, while the pollutant concentration term has a negative sign of the effect, which shows a greater influence on adsorption at a lower level. All three processes were influenced by the interaction between pH and the metal ions' concentration, while the removal Pb(II) and Zn(II) was also influenced by the interaction between $D_{NC}$ and the metal ions' concentration. According to experimental measurements, these interactions between parameters are possible outside of the studied range, as seen in the interaction effect plots. (Figure S1 in the supplementary materials).

The indices $R^2$ and $R^2$adj corresponding to the regression Equations (5)–(7) are close to 1, which denotes a reliable estimate of the metal ion removal efficiency in the BBD space: 99%, 99% and 97% of the variations could be explained by the independent variables for the removal of Cu(II), Pb(II) and Zn(II) ions, respectively. The very small *p*-test value suggests that there is a significant relationship between the variables in the model and that the model fits the data well. The critical values of the Fisher test of the statistical tables according to the degrees of freedom of regression models are below the F-values of Equations (5)–(7) ($F_{(0.05,5,9)}$ = 3.48 for Cu(II), $F_{(0.05,6,8)}$ = 3.86 for Pb(II) and $F_{(0.05,5,9)}$ = 3.48 for Zn(II)), thus highlighting the regression model's high adequacy and significance.

ANOVA results for metal ion removal models are presented in Table 4 and prove that regressions (5)–(7) are statistically significant at a confidence level of 95%. For all three models, the lack of fit of these models is not statistically significant when $p > 0.05$.

**Table 4.** ANOVA for Cu(II), Pb(II), and Zn(II) removal efficiency (%).

| Pollutant | Factor | Df | Sum Sq | Mean Sq | F Value | Pr (>F) |
|-----------|--------|----|--------|---------|---------|---------|
| Cu(II) | $(X_2)^2$ | 1 | 11.2 | 11.23 | 6.0176 | $3.65 \times 10^{-2}$ |
| | $X_1, X_2, X_3$ | 3 | 5678.7 | 1892.89 | 1014.1894 | $1.062 \times 10^{-11}$ |
| | $X_1:X_2$ | 1 | 62.4 | 62.41 | 33.4386 | $2.6 \times 10^{-4}$ |
| | Residuals | 10 | 16.8 | 1.87 | | |
| | Lack of fit | 8 | 16.1 | 2.30 | 6.6491 | $13.69 \times 10^{-2}$ |
| | Pure error | 2 | 0.7 | 0.35 | | |
| Pb(II) | $(X_1)^2$ | 1 | 92.7 | 92.73 | 43.779 | $1.6 \times 10^{-4}$ |
| | $X_1, X_2, X_3$ | 3 | 5551.3 | 1850.43 | 873.645 | $2.11 \times 10^{-10}$ |
| | $X_1:X_2$ | 1 | 25.6 | 25.55 | 12.064 | $8.4 \times 10^{-3}$ |
| | $X_2:X_3$ | 1 | 33.4 | 33.41 | 15.773 | $4.1 \times 10^{-3}$ |
| | Residuals | 8 | 16.9 | 2.12 | | |
| | Lack of fit | 6 | 16.1 | 2.68 | 6.084 | $14.78 \times 10^{-2}$ |
| | Pure error | 2 | 0.9 | 0.44 | | |
| Zn (II) | $X_1, X_2, X_3$ | 3 | 4482.7 | 1494.24 | 90.287 | $4.90 \times 10^{-7}$ |
| | $X_1:X_2$ | 1 | 160.3 | 160.28 | 9.6844 | $1.25 \times 10^{-2}$ |
| | $X_2:X_3$ | 1 | 107.6 | 107.64 | 6.504 | $31.2 \times 10^{-3}$ |
| | Residuals | 9 | 148.9 | 16.55 | | |
| | Lack of fit | 7 | 146.5 | 20.92 | 16.7584 | $5.75 \times 10^{-2}$ |
| | Pure error | 2 | 2.5 | 1.25 | | |

### 3.2.3. Optimization of Metal Ions' Removal

The stationary points of Equations (5)–(7) are located at the edge of or outside the designed experimental space; therefore, a further exploration on the direction of the steepest ascent was needed for model validation or refinement, if that were to be the case. Starting from the stationary points, we estimated the expected increase along the ridge of the steepest ascent and selected two points for each model for additional measurements (Table 5). The predicted data were found to be in good agreement with the experimental determinations, with a standard deviation of 6.37 for Cu(II), 3.29 for Pb(II) and 3.65 for Zn(II); thus, the regression equations are reliable and accurate.

**Table 5.** The predicted and measured removal efficiency in points defining the steepest ascent *t*.

| Pollutant | pH | Initial Concentration (mg/L) | Adsorbent Dose (g/L) | Removal Percentage (%) | |
|---|---|---|---|---|---|
| | | | | Predicted | Experimental |
| Cu(II) | 7.0 | 10 | 1.25 | 83.37 | 82.88 |
| | 6.0 | 100 | 1 | 46.61 | 42.79 |
| | 9.0 | 10 | 1 | 98.10 | 86.96 |
| | 6.5 | 100 | 1.5 | 60.64 | 58.65 |
| | 8.0 | 10 | 0.25 | 69.41 | 61.63 |
| Pb(II) | 7.0 | 10 | 1.25 | 87.54 | 86.66 |
| | 6.0 | 100 | 1 | 59.91 | 57.20 |
| | 9.0 | 10 | 1 | 89.80 | 88.93 |
| | 6.4 | 100 | 1.5 | 70.98 | 65.25 |
| | 8.0 | 10 | 0.25 | 77.08 | 73.54 |
| Zn(II) | 7.0 | 30 | 1.25 | 71.02 | 69.75 |
| | 6.0 | 100 | 1 | 39.12 | 38.90 |
| | 9.0 | 10 | 1 | 82.55 | 78.22 |
| | 6.4 | 100 | 1.5 | 52.13 | 45.62 |
| | 8.0 | 10 | 0.25 | 45.66 | 43.66 |

In order to obtain a better understanding of the influence of each variable, we generated the response surface plots based on the selected regression models. Figure 6 shows the variation of the removal efficiency as a function of the pairing of two independent variables for the removal of Cu(II) (a–c), Pb(II) (d–f) and Zn(II) (g–i) ions.

In Figure 6a,d,g, the removal efficiency of metal ions is represented depending on the initial concentration of the solutions and solution pH due to the initial metal concentration being the factor that imposes the working conditions. A similar influence of the initial metal concentration could also be observed in the case of its combined effect with the dose of the adsorbent, as shown in Figure 6c,f,i. The removal efficiency of Cu(II), Pb(II) and Zn(II) in standard wastewater containing up to 40–50 mg/L metal pollutant is maintained at an increased rate if the adsorbent dose is at least 1 g/L and if the pH of the solution is higher than 6. In the case of heavily concentrated solutions of metals (over 150 mg/L), the efficiency reaches a maximum of 50% even when using the maximum doses of the sorbent or a predominantly basic pH.

The predicted values for the removal efficiencies were positively influenced by working in a basic environment and high adsorbent doses (Figure 6b,e,h). Therefore, to obtain a maximum removal of the investigated metal ions from the solutions, the regression models propose the use of a predominantly basic environment (pH > 8) and a dose of NC of at least 1.5 g/L. Yet, in the case of the metal ions investigated in our study, practice recommends using a pH lower than 7 to avoid the precipitation of metal ions in the solution.

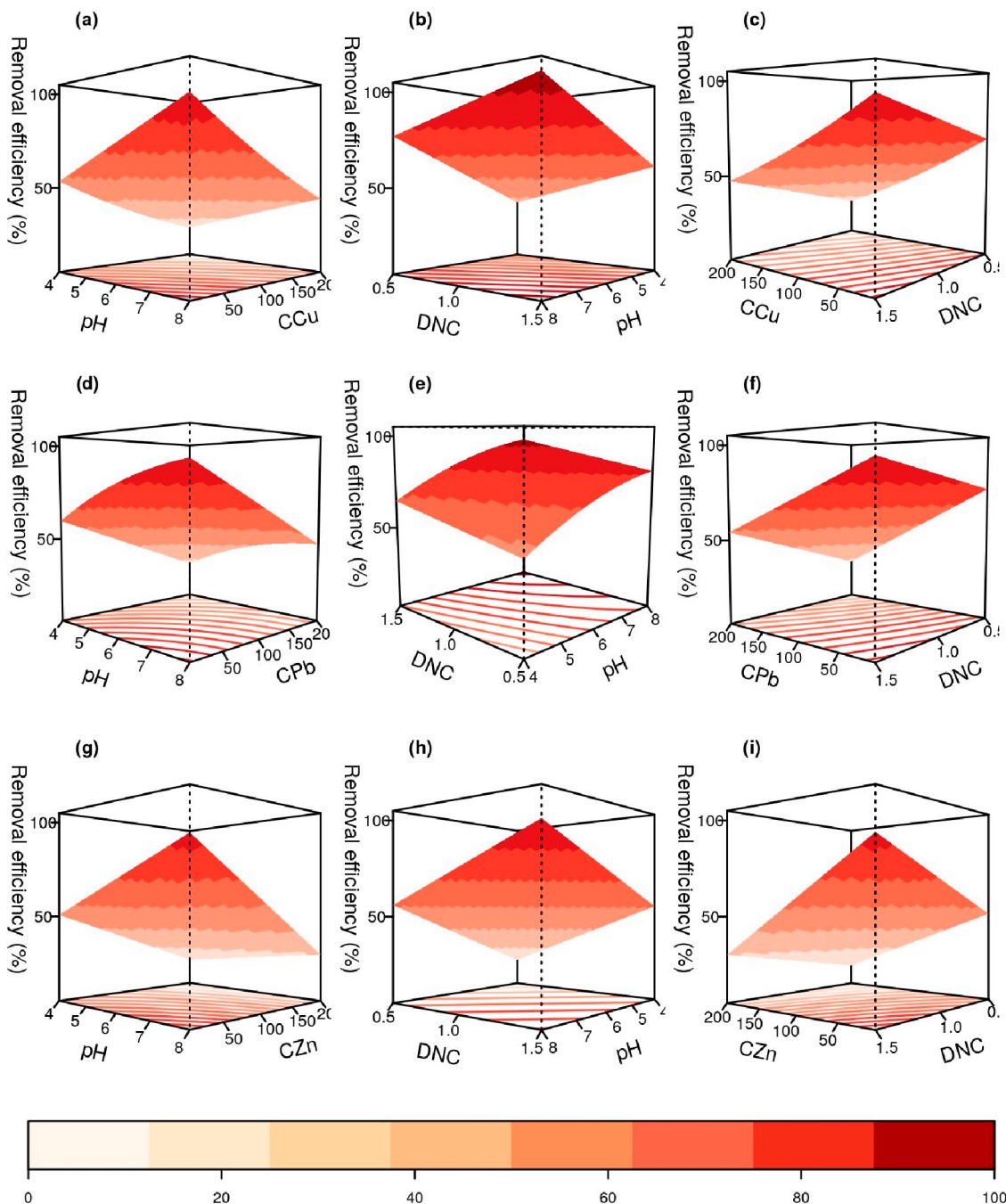

**Figure 6.** Response surface plot for Cu(II): (**a–c**); Pb(II): (**d–f**); and Zn(II): (**g–i**) removal at stationary point slice.

In addition, from an economic and ecological point of view, it is recommended that the adsorption process be carried out at the pH of the solution using the smallest amount of sorbent that ensures maximum efficiency.

With this in consideration, the kinetics and equilibrium adsorption studies carried out in the following sections were performed at the natural pH of the metal ion solution (~5.8), using a NC dose of 1 g/L. It is noteworthy that although the removal efficiency is not maximum when using these conditions, these represent the best combination of working parameters that provides a good removal rate and also obeys all of the previously mentioned issues: the usual concentration of metals in wastewater, the practical recommendations for working with aqueous solutions, and the ecological and economical aspects.

### 3.3. Adsorption Experiments

3.3.1. Influence of Solution pH

The solution pH and the pH$_{PZC}$ of the investigated adsorbent material are very important factors influencing the adsorption efficiency. The pH$_{PZC}$ of the NC nanocomposite was determined [19,34] to be 10 (Figure 7a); therefore, at solution pH below 10, the surface is positively charged, and above 10, the surface is negatively charged.

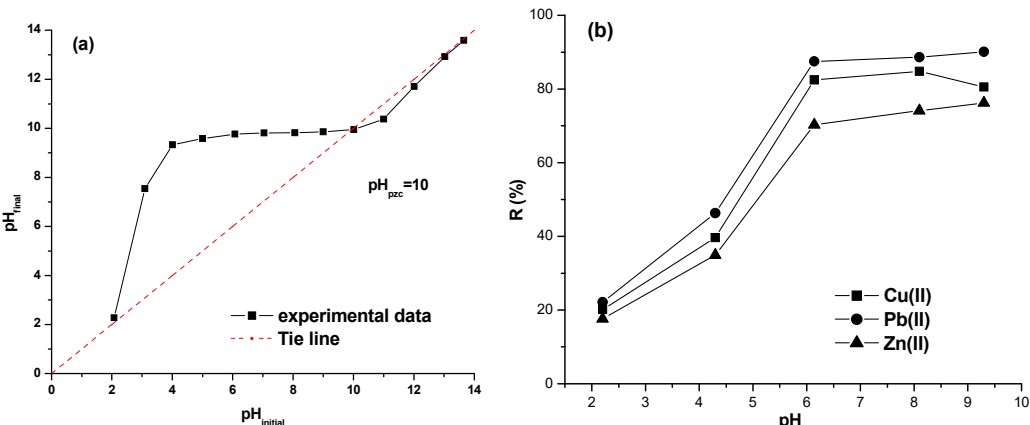

**Figure 7.** (**a**) Point of zero charge (pH$_{PZC}$) of the NC; (**b**) effect of solution pH on adsorption process (1 g/L NC; 10 mg/L metal ions).

As can be seen in Figure 7b, the removal efficiency of metal ions (positively charged) increased as the solution pH increased. Due to electrostatic forces of attraction, at a pH value of the solution above 10, the adsorption efficiency of metal ions should be improved. On the other hand, it is known that at pH 6.5–12.0 for copper, 8.0–8.5 for lead, and 9.0–9.5 for zinc, metal ions can generally be precipitated as a hydroxide in solution [35]. Therefore, in order to avoid the precipitation of copper, lead and zinc ions, the adsorption process should be carried out at a pH value of the solution lower than 7.

Based on the obtained experimental and theoretical results, the experimental kinetic and equilibrium adsorption studies were performed at natural solution pH 5.8 for Cu(II), 5.9 for Pb(II), and 5.8 for Zn(II), respectively.

3.3.2. Kinetics Studies on Adsorption Process

The kinetic studies play a significant role in understanding the dynamics of the adsorption process. In order to minimize the process costs, the experiments were performed at solution pH (5.8 for Cu(II), 5.9 for Pb(II), and 5.8 for Zn(II)), using 1 g/L of NC, at different initial concentrations (10–150 mg/L), and the obtained data were fitted with the linear form of the kinetic models: pseudo-first-order (9) [36], pseudo-second-order (10) [37], and intraparticle diffusion (11) [38]:

$$\log(q_e - q_t) = \log q_e - \frac{k_1}{2.303}t \tag{9}$$

$$\frac{t}{q_t} = \frac{1}{k_2 q_e^2} + \frac{1}{q_e}t \tag{10}$$

$$q_t = k_i t^{0.5} + l \tag{11}$$

in which $q_e$ and $q_t$ represents the amount of pollutant (mg/g) at equilibrium and at any time, $k_1$ represents the rate constant of pseudo-first order adsorption model (min$^{-1}$), $k_2$ the rate constant of pseudo-second order (g/mg min), $k_i$ the rate constant of intraparticle diffusion model, and $l$ represents the thickness of the boundary layer.

By applying Equations (9)–(11) to the experimental data, the values of $k_1$, $k_2$, and $q_e$ were determined from the slope and the interception of the obtained lines (Table 6, Figure S2 from supplementary file).

**Table 6.** Kinetic parameters of pseudo-first-order, pseudo-second-order and intraparticle diffusion equations, applied for adsorption of Cu, Pb and Zn on NC nanocomposite.

| Pollutant | Conc (mg/L) | $q_{e,exp}$ (mg/g) | Pseudo-First-Order Model | | | Pseudo-Second-Order Model | | | Intraparticle Diffusion Model | | |
|---|---|---|---|---|---|---|---|---|---|---|---|
| | | | $q_{e,calc}$ (mg/g) | $k_1 \cdot 10^3$ (min$^{-1}$) | $R^2$ | $q_{e,calc}$ (mg/g) | $k_2 \cdot 10^4$ (g/mg. min) | $R^2$ | $k_i$ (mg/g. min$^{0.5}$) | $l$ | $R^2$ |
| Cu(II) | 10 | 8.25 | 2.79 | 9.05 | 0.7092 | 8.77 | 64.77 | 0.9971 | 0.26 | 4.39 | 0.8955 |
| | 30 | 20.33 | 5.78 | 10.66 | 0.4438 | 20.84 | 16.39 | 0.9976 | 0.32 | 15.39 | 0.9411 |
| | 50 | 31.29 | 9.46 | 18.40 | 0.8985 | 31.60 | 8.39 | 0.9962 | 0.73 | 20.35 | 0.8274 |
| | 100 | 45.22 | 20.83 | 13.91 | 0.5960 | 48.05 | 7.46 | 0.9989 | 0.96 | 32.36 | 0.8782 |
| | 150 | 57.28 | 15.05 | 35.10 | 0.4924 | 53.61 | 7.32 | 0.9914 | 0.73 | 45.65 | 0.6181 |
| Pb(II) | 10 | 8.75 | 4.27 | 18.67 | 0.8608 | 7.11 | 93.71 | 0.9943 | 0.34 | 3.62 | 0.9824 |
| | 30 | 22.41 | 6.76 | 18.91 | 0.7809 | 18.45 | 89.36 | 0.9983 | 0.63 | 12.61 | 0.9775 |
| | 50 | 33.42 | 11.29 | 23.21 | 0.6508 | 28.22 | 82.81 | 0.9984 | 0.97 | 18.57 | 0.9766 |
| | 100 | 57.21 | 15.85 | 15.45 | 0.9299 | 21.93 | 45.93 | 0.9871 | 1.62 | 32.55 | 0.9449 |
| | 150 | 63.45 | 25.47 | 44.10 | 0.9715 | 54.32 | 16.50 | 0.9983 | 2.02 | 32.74 | 0.9461 |
| Zn(II) | 10 | 6.83 | 5.63 | 21.33 | 0.9876 | 7.43 | 62.79 | 0.9989 | 0.21 | 3.75 | 0.9926 |
| | 30 | 17.32 | 10.23 | 14.56 | 0.8139 | 17.69 | 37.68 | 0.9951 | 0.43 | 10.58 | 0.9863 |
| | 50 | 23.96 | 18.83 | 20.46 | 0.9726 | 25.83 | 19.87 | 0.9992 | 0.62 | 14.59 | 0.9825 |
| | 100 | 38.90 | 35.75 | 18.49 | 0.9925 | 38.02 | 14.12 | 0.9977 | 1.15 | 21.31 | 0.9632 |
| | 150 | 43.80 | 31.45 | 21.67 | 0.9421 | 45.47 | 5.95 | 0.9997 | 0.66 | 33.57 | 0.9981 |

The validity of a kinetic model is verified by the correlation coefficient ($R^2$) and the proximity between the calculated adsorption capacity and the experimentally determined value.

Considering these, based on low $R^2$ values (Table 6) it was established that the pseudo-first-order model does not fit well with the experimental data and does not describe the process well.

On the other hand, the high correlation coefficient and the approximation between the calculated and experimental adsorption capacity ($q_e$) prove the applicability of the pseudo-second order model for the characterization of the process. The data obtained show that as the concentration increases, the rate constants decrease, which means that the time required to reach equilibrium will increase, which is in agreement with the experimental results [28,39–41].

Another kinetic model applied was the intraparticle diffusion (Weber–Morris) model (Equation (10)), which is based on diffusive mass transfer and adsorption rate expressed in terms of the square root of time. The values of $k_i$ and $l$ were calculated from the interception and slope of the plot of $q_t$ vs. $t^{0.5}$ and the results are presented in Table 6.

The double linearity plots were obtained, which did not pass through the origin, which indicates that in the adsorption process of Cu(II), Pb(II), and Zn(II) on NC, both the boundary layer and intraparticle diffusion are involved [28,38].

For all of the investigated metal ions, the rate constant for intraparticle diffusion ($k_i$) increased as the concentration increased, in accordance with data obtained by Biswas et al. (2019) [35]. Additionally, the thickness of the boundary layer ($l$) increased with increasing concentration, indicating a higher contribution of surface adsorption in the rate-limiting step [40].

### 3.3.3. Equilibrium Studies on Adsorption Process

Equilibrium studies are significant for the adsorption process as they provide both the value of the maximum adsorption capacity of an adsorbent, as well as explanations of the adsorption mechanism.

In this study, the experimental data obtained working at natural solution pH and room temperature were fitted with the non-linear form of isotherm models: Freundlich, Langmuir, and Redlich–Peterson.

The Freundlich (1907) equation [42] is an empirical relationship between the concentration of a solute adsorbed onto the surface of a solid and the concentration of the solute in the liquid phase.

The Langmuir (1918) [43] equation is a theoretical model that assumes that a fixed number of adsorption sites are available on the surface of solid and, at maximum coverage, there is only a monomolecular layer on the surface.

The Redlich and Peterson (1959) [44] model of the three-parameter model adsorption isotherm combines elements from the Freundlich and Langmuir equations, and the adsorption mechanism does not follow the ideal monomolecular adsorption.

The isotherm equations, the obtained parameters and the statistical parameters applied are presented in Table 7.

**Table 7.** Non-linear adsorption parameters and error parameters of Freundlich, Langmuir and Redlich–Peterson models for copper, lead, and zinc ions adsorption on NC from solutions.

| Isotherm | Equation * | Parameter | Value | | |
|---|---|---|---|---|---|
| | | | Cu(II) | Pb(II) | Zn(II) |
| Freundlich | $q_e = K_F C_e^{1/n}$ | $K_F$ $(mg/g(mg/L)^{-1/n})$ | 10.03 | 10.16 | 5.29 |
| | | $n$ | 2.333 | 2.339 | 2.16 |
| | | $R^2$ | 0.9632 | 0.9634 | 0.9778 |
| | | $\chi^2$ | 26.33 | 26.52 | 6.87 |
| | | AIC | 12.87 | 12.92 | 4.81 |
| | | BIC | 12.46 | 12.50 | 4.40 |
| Langmuir | $q_e = \frac{q_m K_L C_e}{1 + K_L C_e}$ | $q_m$ $(mg/g)$ | 81.36 | 83.54 | 57.11 |
| | | $K_L$ $(L/mg)$ | 0.048 | 0.049 | 0.032 |
| | | $R^2$ | 0.9851 | 0.9833 | 0.9908 |
| | | $\chi^2$ | 11.28 | 12.12 | 2.84 |
| | | AIC | 7.79 | 8.22 | −0.49 |
| | | BIC | 7.37 | 7.80 | −0.90 |
| Redlich–Peterson | $q_e = \frac{K_{RP} C_e}{1 + \alpha_{RP} C_e^\beta}$ | $K_{RP}$ $(L/g)$ | 9.21 | 12.27 | 7.49 |
| | | $\alpha_{RP}$ $(mg/L)^{-\beta}$ | 0.535 | 0.835 | 1.073 |
| | | $\beta$ | 0.677 | 0.643 | 0.585 |
| | | $R^2$ | 0.9728 | 0.9709 | 0.9827 |
| | | $\chi^2$ | 29.19 | 31.57 | 8.04 |
| | | AIC | 15.49 | 15.96 | 7.76 |
| | | BIC | 12.46 | 12.50 | 4.40 |

\* $C_e$ is the metal ion concentration at equilibrium, $q_m$ maximum adsorption capacity of the adsorbent, $K_F$ Freundlich equilibrium constant, $n$ exponent (dimensionless), $K_L$ affinity constant Langmuir, $K_{RP}$, $\alpha_{RP}$ Redlich–Peterson constants, $\beta$ Redlich–Peterson exponent (dimensionless) ($\leq$1).

The parameters of the investigated isotherm models were calculated by non-linear fitting and the models were validated using the statistical measures correlation coefficient

and chi-square. In addition to these statistical parameters, the Akaike information criterion (AIC) [45] and Bayesian information criterion (BIC) [46] were also applied and calculated. (Table 7; Figure S3 from supplementary file).

Based on the high correlation coefficients (0.9851 for Cu(II), 0.9833 for Pb(II) and 0.9908 for Zn(II) ion, respectively) and on the lower chi-square values (11.28 for Cu(II), 12.12 for Pb(II) and 2.84 for Zn(II) ion, respectively), we can appreciate that the adsorption process closely follows the Langmuir model for the investigated metal ions [37,47,48]. Furthermore, the model with the lower AIC and BIC values should be considered the better model to describe the equilibrium adsorption process [46]. The results presented in Table 7 show that the AIC and BIC values for the Langmuir model are the lowest among the three investigated models, while the R2 values are the highest. Therefore, the Langmuir model has the highest reliability. The maximum adsorption capacity value of NC was estimated to be 81.36 mg/g for Cu(II), 83.54 mg/g for Pb(II), and 57.11 mg/g for Zn(II). The values obtained were compared with the published results on the removal of copper, lead and zinc ions using various magnetic nanocomposites, and the data presented in Table 8 revealed the advantages of using NC as an adsorbent.

**Table 8.** Comparison of NC adsorption capacity with different magnetic nanocomposites.

| Pollutant | Magnetic Nanocomposite | Dose (g) | pH | Temp (°C) | $t_e$ (min) | $q_t$ (mg/g) | R (%) | Reference |
|---|---|---|---|---|---|---|---|---|
| Cu (II) | $Fe_3O_4$/talc nanocomposite | 0.12 | - | - | 2 | 100.92 | 72.15 | [47] |
| | Rice straw/magnetic nanocomposites | 0.13 | - | - | 1 | 16.31 | 94.42 | [48] |
| | NiFe2O4/Mod MMT | 0.10 | 6 | 25 | 60 | 18.73 | 99.23 | [49] |
| | magnetite nano-adsorbent | 0.05 | 5.4 | 25 | 120 | 4.41 | 62.61 | [50] |
| | core–shell structured spherical magnetic nanocomposite | 0.03 | - | - | 20 | 35.71 | 80.00 | [51] |
| | Pectin–iron oxide magnetic nanocomposite | 0.02 | 5 | 25 | 1440 | 48.99 | - | [52] |
| | Fe3O4@Carbon nanocomposite | 0.80 | 6.5 | 27 | 120 | 48.08 | 92.00 | [53] |
| | magnetic chitosan/$Al_2O_3$/$Fe_3O_4$ | 1.00 | 5.3 | 15 | 300 | 31.65 | 93.69 | [54] |
| | NC | 1.00 | 5.8 | 25 | 10 | 81.36 | 90.13 | This work |
| Pb(II) | $Fe_3O_4$ magnetite nanoparticles | 0.20 | 5.5 | 25 | 1440 | 37.26 | - | [39] |
| | $Fe_3O_4$/talc nanocomposite | 0.12 | - | - | 2 | 74.62 | 91.35 | [47] |
| | rice straw/magnetic nanocomposites | 0.13 | - | - | 1 | 19.45 | 96.35 | [48] |
| | $Fe_3O_4$@carbon nanocomposite | 0.80 | 6.5 | 27 | 120 | 151.50 | 97.00 | [53] |
| | $Fe_3O_4$@$SiO_2$-NH-pyd | 0.10 | 7 | 25 | 20 | 72.00 | 96.00 | [55] |
| | MNPs35@$SiO_2$ | 0.10 | - | - | 90 | 17.10 | - | [56] |
| | $Fe_3O_4$@MCM-41-$NH_2$ | 0.24 | - | 25 | - | 46.08 | 95.15 | [57] |
| | NPZEO3 | 0.50 | 5.5 | 27 | 5 | 252 | 99.90 | [58] |
| | NC | 1.00 | 5.9 | 25 | 10 | 83.48 | 80.32 | This work |
| Zn(II) | magnetite/carbon nanocomposites | 1.00 | 6.1 | 25 | 240 | 48.45 | 76.90 | [25] |
| | $Fe_3O_4$ magnetite nanoparticles | 0.20 | 5.5 | 25 | 1440 | 9.10 | - | [39] |
| | magnetic nickel ferrite-modified montmorillonite nanocomposite | 0.10 | 6 | 25 | 90 | 5.13 | 91.67 | [47] |
| | magnetic chitosan/$Al_2O_3$/$Fe_3O_4$ | 1.00 | 5.3 | 15 | 300 | 24.27 | 83.81 | [54] |
| | magnetic zeolite/cellulose nanofibers | 0.03 | 7 | 30 | 120 | 9.45 | 96.00 | [59] |
| | alginate-SBA-15 sorbent nanocomposite | 1.00 | - | - | 240 | 46.30 | - | [60] |
| | NC | 1.00 | 5.8 | 25 | 10 | 58.47 | 91.53 | This work |

### 3.3.4. Reusability Study

In order to evaluate the regeneration and reuse of the investigated adsorbent, studies were performed for five adsorption–desorption cycles using 3 g/l NC dose and 10 mg/L metal ions solutions, working at solution pH (5.8 for Cu ions, 5.9 for Pb(II), and 5.8 for Zn(II), respectively) and room temperature. As shown in Figure 8, the loss of removal efficiency from the first to the third cycle is 2.31% for Cu(II) (90.13% to 88.04%), 1.63% for Pb(II) (91.53% to 90.04%), and 2.49% for Zn(II) (from 80.32% to 78.32%). With the continuation of the adsorption–desorption studies, the removal efficiency decreased further until the fifth cycle, which registered a removal efficiency of 17.48% for copper ions, 16.08% for lead ions, and by 21.82% for zinc ions, indicating the saturation of the adsorbent surface.

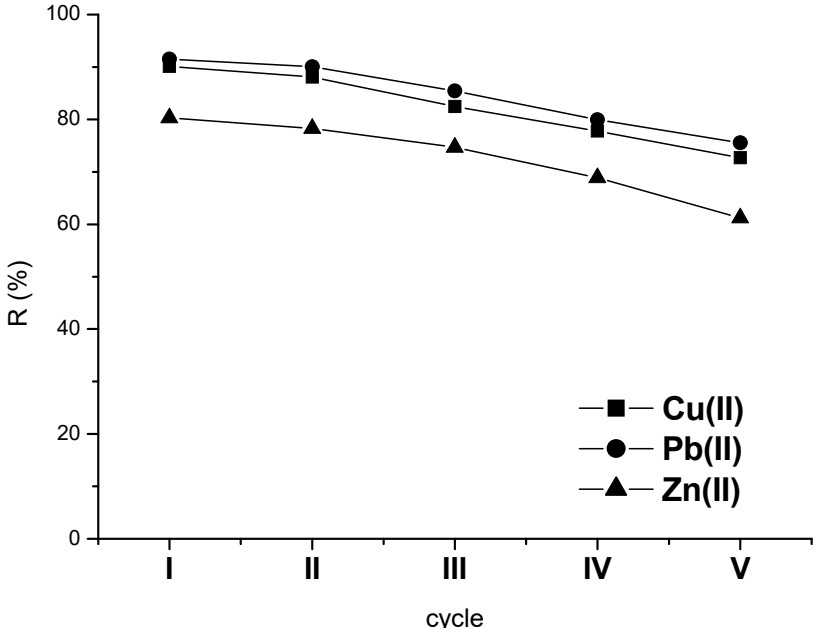

**Figure 8.** Adsorption–desorption cycles: 3 g/L NC, 10 mg/L and pH: 5.8 for Cu(II), 5.9 for Pb(II), and 5.8 for Zn(II) 240 min.

After the five cycles of adsorption–desorption, the removal efficiency was still higher than 70% for copper and lead ions and 60% for zinc ions, and the average removal efficiency for five cycles was 82.21% for Cu(II), 84.50% for Pb(II), and 72.68% for Zn(II). The results obtained show that the magnetic nanocomposite obtained has very good stability, recycling and reuse capacities.

### 4. Conclusions

In this paper, a new carbon magnetic nanocomposite material has been successfully applied as an adsorbent for the removal of copper, lead and zinc ions from aqueous solutions. The specific surface area of NC (755.2 $m^2$/g) being very close to that of carbon (889.9 $m^2$/g) indicates that the obtained nanomaterial has a very good adsorption capacity, and the presence of magnetite in the NC composition guarantees a good and easy separation of phases, which is due to the NC's magnetic properties.

Quadratic and linear models were developed to predict and characterize the relationship between the independent variables and efficiency rate for copper, lead, and zinc ions removal. The significance for each variable on the pollutants' adsorption was established, and the optimal conditions, in accordance with real practice, and advantageous from an economic and ecological perspective, have been recommended. To ensure maximum adsorption efficiency with minimum costs, our investigations were performed at the natural pH of the solution (~5.8) using 1 g/L NC, which generated high removal efficiencies for all of the metal ions investigated: 81.36% Cu(II), 87.46% Pb(II), and 70.30% for Zn(II).

The experimental kinetic data fit well with the second-order kinetic model, and the low values obtained for the rate constants ($k_2$) highlight the fact that the adsorption of Cu(II), Pb(II) and Zn(II) ions is fast.

The maximum adsorption capacities of the NC nanocomposite obtained from the Langmuir model were as follows: 81.36 mg/g for the removal of copper ions, 83.54 mg/g for lead ions, and 57.11 mg/g for zinc ions.

Obtaining a removal efficiency greater than 70% after five consecutive cycles of adsorption–desorption highlights the good regeneration and reuse of the investigated nanocomposite. The results obtained in this study indicate that NC can be applied as an effective adsorbent for the removal of metal ions from wastewater.

**Supplementary Materials:** The following supporting information can be downloaded at: https://www.mdpi.com/article/10.3390/magnetochemistry9070163/s1, Figure S1: Interaction effect plot for Cu(II) (a–c), Pb(II) (d–f), and Zn(II) (g–i) ions removal; Figure S2: Plots of: pseudo-first kinetic model for the adsorption of Cu(II) (a), Zn(II) (d), Pb(II) (g); pseudo-second kinetic model for the adsorption of Cu(II) (b), Zn(II) (e), Pb(II) (h); intraparticle diffusion model for the adsorption of Cu(II) (c), Zn(II) (f), Pb(II) (i)on NC; Figure S3: Isotherm plots for the adsorption of Cu(II) (a), Pb(II) (b), and Zn(II) (c) on NC.

**Author Contributions:** S.G.M.: conceptualization, methodology, supervision, writing—original draft, review and editing, L.H.: formal analysis, software, methodology, writing—original draft, review and editing. M.A.N.: investigation (synthesis, characterization, adsorption studies), formal analysis. C.P.: supervision, validation, visualization. All authors have read and agreed to the published version of the manuscript.

**Funding:** This research received no external funding.

**Institutional Review Board Statement:** Not applicable.

**Informed Consent Statement:** Not applicable.

**Data Availability Statement:** The data presented in this study are available on request from the corresponding authors.

**Acknowledgments:** This work was supported by Research Projects 2.3 "Coriolan Dragulescu" Institute of Chemistry.

**Conflicts of Interest:** The authors declare that they have no known competing financial interest or personal relationships that could appear to have influenced the work reported in this paper.

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
