# Peer review of "Removal of Metal Ions via Adsorption Using Carbon Magnetic Nanocomposites: Optimization through Response Surface Methodology, Kinetic and Thermodynamic Studies"

_magnetochemistry, doi:10.3390/magnetochemistry9070163_

Round 1
Reviewer 1 Report
I enjoyed the provided manuscript, and I strongly support it acceptance for publication. However, I have several minor remarks on the text and also several suggestions about additional experiments which can make the paper stronger, in my opinion. I cannot estimate the possible timing to include these additional results (it depends on the original material availability and the access to the equipment) - therefore, please take the suggestions as kind advices rather than the ultimate need.
1. Please correct unnecessary capitalization of Lead, Zinc, and Copper throughout the text.
2. In line 12, 'simulated wastewaters' seem strange to me. Maybe not incorrect, but... Please consider something like "artificial mixtures", "model mixtures" or whatever. "Simulated wastewaters" sounds like a wastewater was simulated using some computer software. Probably I am wrong - just a suggestion.
3. In line 51 it is stated that activated carbon is "effective but expensive adsorbent material". This seems strange to me, because I have considered carbon among the cheapest materials. Please explain.
4. In lines 88-91 - please repeat the procedure at least briefly. The reference to the original publication is ok but the synthesis should be understood by the readers without consulting external sources. Also, please explain briefly the role of each of the components in the mixture. It is stated somewhere in the text that citric acid is a "fuel", but the role of silver nitrate in the mixture and this of silver in the composite has never been mentioned. In ref. 19 it was mentioned that Ag was introduced in the composite for its antibacterial and catalytic properties - but they are not so relevant to heavy metals removal. Is silver in the composite essential in the context of this study?
5. Please remove molar masses in lines 92-93. This is very basic reference information, and it seems useless here.
6. In line 116, ref. 30 seems very strange and not appropriate. In this place, a link to the equipment manufacture could be placed. However, ref. 30 is a completely different study, the description of the magnetization experiment in which just takes 7 lines and refers to a further paper:
"The basic magnetic
properties of the two oxides were examined in low frequency
(50 Hz) ac magnetic fields, by means of a conventional induction
hysteresigraph [21], calibrated to a high purity (99.98%) Ni
reference; the output signals (corresponding to the time dependent
field and magnetization) of the device were recorded in
numeric format (ASCII files) to a PC, by means of the DT-9816 A
(Data Translation Inc.) data acquisition board (16 bit analog-todigital
conversion, parallel inputs)."
If the authors of the present study feel that the description is so important (for me it is not - it is just one of the standard characterization methods) - it is better to include the explicit description rather than this reference.
7. In lines 166-167 - please indicate whether molar or mass fractions of the components are reported.
8. Regarding the data in Figs. 1 and 3 - it would be nice to calculate the crystallites size from the XRD pattern (via the Scherrer equation, for example) and compare this with the particles size from SEM. Do the authors have the EDX mapping of the elements? Do the author have the information of the initial and final size of the carbon particles; it it changed during the synthesis? Is the carbon surface partially oxidized (this can be verified from the IR spectra)? In my view, these are important issues regarding the synthesis method. If this has been earlier reported elsewhere - please give these descriptions with the reference.
9. Line 189 - the IUPAC classification should be properly referenced.
10. In lines 223-224 it is stated that "The variation line of this parameter has the highest steepness, therefore it seems to influence the most the removal efficiency. " However, it is clear that the line steepness depends on the level of the actual parameter change. The change in the coded variable by 2 units corresponds to 3x change in the adsorbent concentration and 20x change in the metals concentration. For me it is no surprise then that the metal concentration gives the steepest line in the coded scale... I would suggest removing of this statement, saying instead that the considered variables were found significant.
11. Regarding Fig. 5 - I would add the SE limits to confirm the effects significance. And why the points for level 0 of the variables are not shown?
12. I do not like Fig. 6 and the related discussion. I am more used to a reverse order of the discussion - first building the model, then finding that some of the interactions are significant and then demonstrating this in the interaction plots. I would suggest at least put Fig. 6 to supplementary part, not in the main text. At most, it the authors agree with my view, the discussion can be somewhat modified to mention these plots after the models.
13. Please report the DF values in Table 3 along with the corresponding F-values. Also, I would suggest decreasing the number of the digits in the tabulated parameters, just to make it clearer.
14. In lines 263-264 it is stated that "A very good fit of the experi-263 mental results with equations (5-7) is illustrated by the extremely low value of the p-value 264 of each model. "
For me, this statement seems wrong. The low p-value means that the model is OVERALL statistically significant, i.e. at least one of the terms (besides the intercept) is nonzero. Just a week ago processing of my own data gave R2 of 0.4 and high MSe (extremely poor fit) and the model p-value of 1e-10.
The good fit of the provided models is confirmed by high R2 values, not low p-value or high F-value, I suppose.
By the way, have you tried cross-validation of the models to confirm the absence of overfitting? This is of course confirmed by the data in Table 5 - but additional proof would be also nice.
15. Lines 279-280 - it is better to report the mean deviation of the calculated values from the experimental ones. The values given in the text (96 and 99%) can be easily messed up with the actual degree of removal, also measured in %...
16. Fig. 7 - please add the scale explaining the colors (or at least describe them in the figure caption).
17. Fig. 8 (caption): "MSC" is not explained in the text.
18. Regarding discussion of the data in Fig. 8: it is well explained by the authors why the removal degree is increased with increasing pH. But what is the driving force of the adsorption at pH < PZC where the surface is positively charged?
19. Please provide the references where eq. 9-11 are discussed in more detail. This will be instructive to the readers who are relatively new to this field.
20. Regarding discussion of Table 6 - I would suggest adding some plots with the data points and the fitted lines to illustrate the quality of the fitting equations.
21. Line 378: the Sips isotherm is mentioned here but is never discussed in the further text. It is also announced in the Table 7 caption (line 391), but the data are not provided.
22. Same as i. 20 - discussion of Table 7 should also be supported by the fitted plots, in my opinion. This will also show the metal concentrations range and the number of data points, which are important in the context of the isotherm models fitting.
23. Again regarding Table 7. As far as I can understand, at KRP = qmKL, beta = 1, and alphaRP = KL the Redlich-Peterson equation is reduced to the Langmuir equation. Hence, at these values of the RP parameters the corresponding RP equation should fit the data at least as good as the L equation. If so, why the R2 parameter of the RP equation is lower than this for the L equation? Even if these are adjusted R2, I do not think that the more general equation (PR) should be poorer than its special case (L).
In general, English is fine. I have spotted several mistakes but they do not kill the text. However, if the authors find an opportunity to share the manuscript with a native English-speaking colleague, he/she can probably make corrections to make the text smoother.
Author Response
I enjoyed the provided manuscript, and I strongly support it acceptance for publication. However, I have several minor remarks on the text and also several suggestions about additional experiments which can make the paper stronger, in my opinion. I cannot estimate the possible timing to include these additional results (it depends on the original material availability and the access to the equipment) - therefore, please take the suggestions as kind advices rather than the ultimate need.
We thank the reviewer for appreciating our work and the results presented in the paper
- Please correct unnecessary capitalization of Lead, Zinc, and Copper throughout the text.
Answer: Thank you for suggestion. We made the correction in the revised manuscript
- In line 12, 'simulated wastewaters' seem strange to me. Maybe not incorrect, but... Please consider something like "artificial mixtures", "model mixtures" or whatever. "Simulated wastewaters" sounds like a wastewater was simulated using some computer software. Probably I am wrong - just a suggestion.
Answer: We replace “simulated wastewaters' with “aqueous solutions” in the revised manuscript
- In line 51 it is stated that activated carbon is "effective but expensive adsorbent material". This seems strange to me, because I have considered carbon among the cheapest materials. Please explain.
Answer: it was a mistake of expression. Indeed, activated carbon is a low-cost material, but we wanted to refer to the fact that the manufacture of adsorbents of this type requires a high cost considering the origin of the raw material.
Activated carbon is one of the most popular adsorbents, widely used in wastewater treatment, but which presents numerous disadvantages: high production costs, regeneration problems, low mechanical properties and difficulties in separating the liquid-solid phases.
We rephrase in the revised manuscript: “After extensive use in wastewater treatment, activated carbon has proven to be an effective absorbent material, but it presents many disadvantages such as: high production costs, low mechanical properties and most importantly its difficult separation from the solution, which causes problems of regeneration and reuse”.
- In lines 88-91 - please repeat the procedure at least briefly. The reference to the original publication is ok but the synthesis should be understood by the readers without consulting external sources. Also, please explain briefly the role of each of the components in the mixture. It is stated somewhere in the text that citric acid is a "fuel", but the role of silver nitrate in the mixture and this of silver in the composite has never been mentioned. In ref. 19 it was mentioned that Ag was introduced in the composite for its antibacterial and catalytic properties - but they are not so relevant to heavy metals removal. Is silver in the composite essential in the context of this study?
Answer: For the synthesis of NC, iron nitrate respectively silver nitrate were used as oxidizing agents, and citric acid were used as fuels (reducing agents). In order not to be accused of self-plagiarism, we briefly described the synthesis procedure. It is described in detail in a previous work.
We have used Ag considering the well known catalytic and antibacterial properties of silver which makes possible the use of this nanocomposite in various fields.
There are many reports in the literature that highlight the good adsorption capacity of silver nanoparticles through a mechanism involving electrostatic attraction forces [1-3].
Therefore, the inclusion of silver in the NC nanocomposite aimed to:
- Increasing the adsorption capacity of the adsorbent by its presence alongside carbon in the nanocomposite.
- Considering that the pHPZC value for silver is 8.6 (according to the data from the literature [4]) as in the case of maghemite, the aim was the increase of the electrostatic attraction forces between the surface of the adsorbent and that of the metal ions.
- Also, knowing the antibacterial properties of silver, we will investigate this property of the adsorbent in a future paper.
- Ali, A. Mannan, I. Hussain, I. Hussain, M. Zia, Effective removal of metal ions from aquous solution by silver and zinc nanoparticles functionalized cellulose: Isotherm, kinetics and statistical supposition of process, Environ. Nanotechnol. Monitor. Manag., 9 (2018), Pages 1-11, doi.org/10.1016/j.enmm.2017.11.003
- Kamran Dastafkan, Mostafa Khajeh, Mousa Bohlooli, Mansour Ghaffari-Moghaddam and Nader Sheibani, Mechanism and behavior of silver nanoparticles in aqueous medium as adsorbent, Talanta, http://dx.doi.org/10.1016/j.talanta.2015.03.065
- Mona A .Darweesh, Mahmoud Y.Elgendy, Mohamed I. Ayad, Abdel Monem M. Ahmed, N.M.Kamel Elsayed, W.A.Hammad, A unique, inexpensive, and abundantly available adsorbent: composite of synthesized silver nanoparticles (AgNPs) and banana leaves powder (BLP), Heliyon, 8(4), 2022, e09279, https://doi.org/10.1016/j.heliyon.2022.e09279
- Folasegun A. Dawodu, Courage U. Onuh, Kovo Akpomie, Emmanuel Iyayi Unuabonah Synthesis of silver nanoparticle from Vigna unguiculata stem as adsorbent for malachite green in a batch system, Springer, SN Applied Sciences 1(4), April 2019 DOI: 10.1007/s42452-019-0353-3
- Please remove molar masses in lines 92-93. This is very basic reference information, and it seems useless here.
Answer: Thank you for suggestion. We remove molar masses in the revised manuscript
- In line 116, ref. 30 seems very strange and not appropriate. In this place, a link to the equipment manufacture could be placed. However, ref. 30 is a completely different study, the description of the magnetization experiment in which just takes 7 lines and refers to a further paper:
"The basic magnetic properties of the two oxides were examined in low frequency (50 Hz) ac magnetic fields, by means of a conventional induction hysteresigraph [21], calibrated to a high purity (99.98%) Ni reference; the output signals (corresponding to the time dependent field and magnetization) of the device were recorded in numeric format (ASCII files) to a PC, by means of the DT-9816 A (Data Translation Inc.) data acquisition board (16 bit analog-todigital conversion, parallel inputs)."
If the authors of the present study feel that the description is so important (for me it is not - it is just one of the standard characterization methods) - it is better to include the explicit description rather than this reference.
Answer: At the reviewer suggestion we have removed the reference [30].
- In lines 166-167 - please indicate whether molar or mass fractions of the components are reported.
Answer: In lines 166-167 are reported mass percent of the components. At the reviewer suggestion we have indicated this in the manuscript
- Regarding the data in Figs. 1 and 3 - it would be nice to calculate the crystallites size from the XRD pattern (via the Scherrer equation, for example) and compare this with the particles size from SEM. Do the authors have the EDX mapping of the elements? Do the author have the information of the initial and final size of the carbon particles; it it changed during the synthesis? Is the carbon surface partially oxidized (this can be verified from the IR spectra)? In my view, these are important issues regarding the synthesis method. If this has been earlier reported elsewhere - please give these descriptions with the reference.
Answer: The aim of the work is to demonstrate the advantages of using the nanocomposite both in terms of its adsorption capacity and the easy separation of phases with the help of a magnet.
We did not compare the diameter of the crystallites with that of the particles because the fact that the nanocomposite is monocrystalline (if the diameter of the crystallites is equal to the diameter of the particles) or is polycrystalline (if the diameter of the crystallites is smaller than that of the particles) does not influence the adsorption capacity of the nanocomposite.
We did not perform the EDX mapping of the elements because this does not provide additional information regarding the adsorption capacity of the nanocomposite.
We have no information on the initial and final size of the carbon particles but we have the specific surface area of pure carbon (889.9 m2/g) and that of the nanocomposite (755.2 m2/g), which represents the most important characteristic of an adsorbent. The higher the specific surface area, the better the adsorption capacity.
Oxidation of the carbon surface cannot occur during the studied adsorption processes. Oxidation of carbon with the release of CO and/or CO2 can only occur thermally at about 600°C, as it follows from the thermal analysis (Fig. 2) or by chemical treatment, for example with H2O2, as we demonstrated in the works:
- IanoÅŸ, R. Lazău, I. Lazău, C. Păcurariu, Chemical oxidation of residual carbon from ZnAl2O4 powders prepared by combustion synthesis, Journal of the European Ceramic Society, 32, 2012, 1605–1611.
- IanoÈ™, E.A. Moacă, A. Căpraru, R. Lazău, C. Păcurariu, Maghemite, γ-Fe2O3, nanoparticles preparation via carbon-templated solution combustion synthesis, Ceramics International, 44(12), 2018, 14090-14094.
Under the conditions of the adsorption processes studied by us (room temperature, lack of oxidizing agents), the carbon oxidation is impossible to occur.
- Line 189 - the IUPAC classification should be properly referenced.
Answer: The IUPAC classification has been properly referenced
- In lines 223-224 it is stated that "The variation line of this parameter has the highest steepness, therefore it seems to influence the most the removal efficiency. " However, it is clear that the line steepness depends on the level of the actual parameter change. The change in the coded variable by 2 units corresponds to 3x change in the adsorbent concentration and 20x change in the metals concentration. For me it is no surprise then that the metal concentration gives the steepest line in the coded scale... I would suggest removing of this statement, saying instead that the considered variables were found significant.
Answer: The statement was removed in the revised manuscript. We add another sentence saying that the parameters were found significant for removal efficiency.
- Regarding Fig. 5 - I would add the SE limits to confirm the effects significance. And why the points for level 0 of the variables are not shown?
Answer: We appreciate the referent’s suggestions and we have redone Figure 5 accordingly, adding the points for level 0 of the variables. Regarding the representation of the standard deviation, we think it is not justified in our case. The average value for the points in the minimum and maximum level is based on only 4 values, and only the level 0 points were replicated. We have mentioned the standard deviation for level 0 in the text, lines 217-218. We thank the referent for the idea and we will consider designing the experiment with 2 replications in future studies.
- I do not like Fig. 6 and the related discussion. I am more used to a reverse order of the discussion - first building the model, then finding that some of the interactions are significant and then demonstrating this in the interaction plots. I would suggest at least put Fig. 6 to supplementary part, not in the main text. At most, it the authors agree with my view, the discussion can be somewhat modified to mention these plots after the models.
Answer: Figure 6 has been moved to supplementary materials (Figure S1), section 3.2.1. and the following figures were renamed and renumbered accordingly. Also, the discussion related to this figure was reduced to a sentence, after the presentation of the models (lines 263-265).
- Please report the DF values in Table 3 along with the corresponding F-values. Also, I would suggest decreasing the number of the digits in the tabulated parameters, just to make it clearer.
Answer: We decreased the number of the digits in Table 3. The Df values ​​for the regressions were included in Table 4. Also, the critical F values ​​for df1 and df2 related to each regression, are given in the text, line 274
- In lines 263-264 it is stated that "A very good fit of the experi-263 mental results with equations (5-7) is illustrated by the extremely low value of the p-value 264 of each model".
For me, this statement seems wrong. The low p-value means that the model is OVERALL statistically significant, i.e. at least one of the terms (besides the intercept) is nonzero. Just a week ago processing of my own data gave R2 of 0.4 and high MSe (extremely poor fit) and the model p-value of 1e-10.
The good fit of the provided models is confirmed by high R2 values, not low p-value or high F-value, I suppose.
Answer: We thank the referent for the observation; the sentence has been adjusted accordingly
By the way, have you tried cross-validation of the models to confirm the absence of overfitting? This is of course confirmed by the data in Table 5 - but additional proof would be also nice.
Answer: The initial measurements used to develop the regression models were selected based on a factorial design that involves exploring the response space with a minimum number of experiments. Therefore, each of the 12+1 (center) selected points ensures representativeness for an area of ​​the response space. Cross-validation involves dividing the data into learning and testing sets, therefore it is very possible to lose the information provided by each experimental point and to reduce the quality of the model estimation. For these reasons, in this particular case (factorial design) we choose only the experimental validation, which we consider to be a direct and relevant form of verification of the adequacy of the model to reality. At the referent's suggestion, we performed a cross-validation, using the LOO method. The results are below for 15 samples and 3 predictors.
Metal ions |
Rsquared |
RMSE |
MAE |
[Range of actual values] ​​(mean) |
Actual error related to the average value of the removal efficiency |
Cu |
0.989931 |
1.986048 |
1.626599 |
[16.98-84.79] (46.96) |
0.93 |
Pb |
0.9866967 |
2.278796 |
1.759222 |
[18.72-88.65] (53.42) |
1.31 |
Zn |
0.8952207 |
5.905015 |
4.641728 |
[10.12-74.14] (36.76) |
2.16 |
According to the Rsquared values, the models explain well the variation of the real values ​​of the dependent variable. Also, RMSE or MAE, are within acceptable limits for the variation interval of the data.
- Lines 279-280 - it is better to report the mean deviation of the calculated values from the experimental ones. The values given in the text (96 and 99%) can be easily messed up with the actual degree of removal, also measured in %...
Answer: We thank the referent for this observation. We replaced the percentage values ​​with the calculated values ​​of the standard deviations
- Fig. 7 - please add the scale explaining the colors (or at least describe them in the figure caption).
Answer: We added a legend with the color scale for Figure 7 (Now Figure 6)
- Fig. 8 (caption): "MSC" is not explained in the text.
Answer: It was a typing error. We made the correction in the revised manuscript
- Regarding discussion of the data in Fig. 8: it is well explained by the authors why the removal degree is increased with increasing pH. But what is the driving force of the adsorption at pH < PZC where the surface is positively charged?
Answer: The low level of adsorption at low pH values (pH<pHPZC) may be due to the electrostatic forces of repulsion between the positively charged NC surface and the predominant metal species with a positive charge (Cu2+, Pb2+ and Zn2+), and also to the competition between H+ and metal ions for available adsorption centers on the surface of the adsorbent.
- Please provide the references where eq. 9-11 are discussed in more detail. This will be instructive to the readers who are relatively new to this field.
Answer: We have introduced the corresponding references in the revised form:
Lagergren, S. Zur theorie der sogenannten adsorption geloster stoffe. K.-Sven. Vetenskapsakademiens Handl. 1898, 24, 1–39.
Ho, Y., McKay, G. Pseudo-second order model for sorption processes. Process. Biochem. 1999, 34, 451–465.
Wu, F.-C., Tseng, R.-L., Juang, R.-S. Initial behavior of intraparticle diffusion model used in the description of adsorption kinetics. Chem. Eng. J. 2009 153(1-3), 1–8.
- Regarding discussion of Table 6 - I would suggest adding some plots with the data points and the fitted lines to illustrate the quality of the fitting equations.
Answer: We have introduced the kinetic plots in the Supplementary File (Fig. S2).
- Line 378: the Sips isotherm is mentioned here but is never discussed in the further text. It is also announced in the Table 7 caption (line 391), but the data are not provided.
Answer: It was a typing error. We made the correction in the revised manuscript
- Same as i. 20 - discussion of Table 7 should also be supported by the fitted plots, in my opinion. This will also show the metal concentrations range and the number of data points, which are important in the context of the isotherm models fitting.
Answer: We have introduced the isotherm plots in the Supplementary File (Fig. S3).
- Again regarding Table 7. As far as I can understand, at KRP= qmKL, beta = 1, and alphaRP= KL the Redlich-Peterson equation is reduced to the Langmuir equation. Hence, at these values of the RP parameters the corresponding RP equation should fit the data at least as good as the L equation. If so, why the R2 parameter of the RP equation is lower than this for the L equation? Even if these are adjusted R2, I do not think that the more general equation (PR) should be poorer than its special case (L).
Answer: The analysis of the experimental data and determination of the parameters which describe the theoretical models were performed using the Origin 6.1 program. As evaluation criteria: the square multiple regression coefficient (R2) and the chi-square analysis (c2) were used.
As you mentioned only in the limit situation (b = 1) the Redlich-Peterson equation is reduced to the Langmuir equation.
In our case, we are not in the limit situation, b is different from 1, therefore for modeling a regression is used that involves a different statistic (different values for R2 and chi). Nevertheless, as can be seen in Table 7, the differences between the values of R2 are not large in the case of applying the Langmuir and R-P models, respectively.
On the other hand, the model with the lower BIC value should be considered the better model to describe the equilibrium adsorption process. The results in Table 7 show that the values of the c2, AIC and BIC for the Langmuir model are the smallest among the three investigated models, while the values of R2 are the largest. Therefore, the Langmuir model has the highest reliability.
Comments on the Quality of English Language
In general, English is fine. I have spotted several mistakes but they do not kill the text. However, if the authors find an opportunity to share the manuscript with a native English-speaking colleague, he/she can probably make corrections to make the text smoother.
Answer: Thank you for the observation. The English was checked and corrected.
Reviewer 2 Report
After reading the manuscript presented by Muntean et al. I have found some major revisions that must be followed, they are given enumerated:
1) Units must be given for Eq.(1), usually mg g−1.
2) Miller indexes are not given in the diffractogram of Figure 1.
3) I have a major comment here, it is not possible to differentiate pure nanomagnetite and nanomaghemite without rigorous magnetic and surface characterization. A more detail paragraph must be added to clear mention this difference in line 163, I suggest adding these references to reinforce your discussions: (1) Differentiating Nanomaghemite and Nanomagnetite and Discussing Their Importance in Arsenic and Lead Removal from Contaminated Effluents: A Critical Review; (2) Progress toward Room-Temperature Synthesis and Functionalization of Iron-Oxide Nanoparticles
4) Change the word pattern by diffractogram in the whole text, pattern is a high-resolution standard measurement often citable with a crystallographic data base, authors are reporting a relative measurement that has amorphous characteristics and high background.
5) Figure 3 is blurred, not scale is seen in the image (left side) and letter represents the elemental analysis must be increased in size.
6) In lines 201 to 207, the authors found novel results on how a low saturation magnetization (Ms) system can be related to the presence of the carbon matrix, this can be contrasted with these references: (1)Superspinglass state in functionalized zeolite 5A-maghemite nanoparticles; (2)Rietveld Refinement, μ-Raman, X-ray Photoelectron, and Mössbauer Studies of Metal Oxide-Nanoparticles Growth on Multiwall Carbon Nanotubes and Graphene Oxide; (3)Surface magnetic properties of a ternary nanocomposite and its ecotoxicological properties in Daphnia magna; where the reduction of the Ms value was observed in nanocomposite systems. A paragraph highlighting this issue based on acceptable percentage of Ms for environmental applications must be added.
7) Linear kinetic models often underestimate the kinetic parameters, it is preferable to used non-linear kinetic models, this section 3.3.2 must be corrected. Models must be correlated using BIC parameter, please use the next reference for guideline: (1) Enhanced Removal of As(V) and Pb(II) from Drinking and Irrigating Water Effluents Using Hydrothermally Synthesized Zeolite 5A. R2 value only analyze the accuracy of each model independently, hence statistically speaking is a wrong analysis.
8) BIC parameter must be also employed to compare isotherm models in section 3.3.3. Table 7 must be updated with new calculations.
9) Table 8 is incomplete; the first pollutant is missing. More parent system need to be added based on magnetic nanohybrids and parent mesoporous systems: Add these references to compare: (1) Enhanced Removal of As (V) and Pb (II) from Drinking and Irrigating Water Effluents Using Hydrothermally Synthesized Zeolite 5A; (2) Surface Adsorption Mechanism between Lead (II, IV) and Nanomaghemite Studied on Polluted Water Samples Collected from the Peruvian Rivers Mantaro and Cumbaza; (3) Improved removal capacity and equilibrium time of maghemite nanoparticles growth in zeolite type 5A for Pb (II) adsorption. Table 8 must compare kinetic parameters, adsorbent dose, adsorbent size, optimum pH, equilibrium time, removal percentage, temperature, and other important adsorption parameters. A paragraph discussing and comparing must be added.
10) The conclusions section must be written again, it is usually one paragraph reflecting the most important findings.
Other minor points:
-Punctuation, source letter size, and grammar typos must be corrected.
-Increase the resolution of Figure 9.
-Punctuation, source letter size, and grammar typos must be corrected.
Author Response
REVIEWER 2
Responces
After reading the manuscript presented by Muntean et al. I have found some major revisions that must be followed, they are given enumerated:
1) Units must be given for Eq.(1), usually mg g−1.
Answer: Thank you for the observation. We introduced the units in Eq. 1
2) Miller indexes are not given in the diffractogram of Figure 1.
Answer: As demonstrated in the work, the nanocomposite is a mixture of crystalline phases: magnetite, maghemite and silver. Considering the purpose of the paper, the specification of the Miller indices has no relevance. At the reviewer's suggestion, for each separate phase in fig. 1 have also presented Miller's clues.
3) I have a major comment here, it is not possible to differentiate pure nanomagnetite and nanomaghemite without rigorous magnetic and surface characterization. A more detail paragraph must be added to clear mention this difference in line 163, I suggest adding these references to reinforce your discussions: (1) Differentiating Nanomaghemite and Nanomagnetite and Discussing Their Importance in Arsenic and Lead Removal from Contaminated Effluents: A Critical Review; (2) Progress toward Room-Temperature Synthesis and Functionalization of Iron-Oxide Nanoparticles
Answer: As we stated in the paper, it is difficult to distinguish between magnetite and maghemite. This can be done by Mossbauer spectroscopy or X-ray photoelectron spectroscopy. Considering the purpose of the work, it is important to demonstrate that the nanocomposite has magnetic properties regardless of whether they are due to magnetite or maghemite. That is why we consider that the results obtained based on the X-ray diffraction pattern by the PDXL2 software using the relative intensity ratio (RIR) method are sufficient.
4) Change the word pattern by diffractogram in the whole text, pattern is a high-resolution standard measurement often citable with a crystallographic data base, authors are reporting a relative measurement that has amorphous characteristics and high background.
Answer: In all the works consulted by us, published in the most prestigious journals, the name is used:
X-ray diffraction pattern and not diffractogram as suggested by the reviewer.
Some examples are:
- Ji-chun Qu, Cui-ling Ren, Ya-lei Dong, Yan-ping Chang, Min Zhou, Xing-guo Chen, Facile synthesis of multifunctional graphene oxide/AgNPs-Fe3O4 nanocomposite: A highly integrated catalysts, Chemical Engineering Journal, Volumes 211–212, 15 November 2012, Pages 412-420 (F.-16.744)
- Lunhong Ai, Chunying Zhang, Zhonglan Chen, Removal of methylene blue from aqueous solution by a solvothermal-synthesized graphene/magnetite composite, Journal of Hazardous Materials, Volume 192, Issue 3, 15 September 2011, Pages 1515-1524 (F.-14.224)
- Song Bai, Xiaoping Shen, Xin Zhong, Yang Liu, Guoxing Zhu, Xiang Xu, Kangmin Chen, One-pot solvothermal preparation of magnetic reduced graphene oxide-ferrite hybrids for organic dye removal, Carbon, Volume 50, Issue 6, May 2012, Pages 2337-2346 (F.-11.307)
For this reason, we consider that it is not necessary to change the name as suggested by the reviewer.
5) Figure 3 is blurred, not scale is seen in the image (left side) and letter represents the elemental analysis must be increased in size.
Answer: Figure 3 has been resized.
6) In lines 201 to 207, the authors found novel results on how a low saturation magnetization (Ms) system can be related to the presence of the carbon matrix, this can be contrasted with these references: (1)Superspinglass state in functionalized zeolite 5A-maghemite nanoparticles; (2)Rietveld Refinement, μ-Raman, X-ray Photoelectron, and Mössbauer Studies of Metal Oxide-Nanoparticles Growth on Multiwall Carbon Nanotubes and Graphene Oxide; (3)Surface magnetic properties of a ternary nanocomposite and its ecotoxicological properties in Daphnia magna; where the reduction of the Ms value was observed in nanocomposite systems. A paragraph highlighting this issue based on acceptable percentage of Ms for environmental applications must be added.
Answer: As we mentioned before, considering the purpose of the work, it is important to demonstrate that the nanocomposite has magnetic properties. For this reason, we established the saturation magnetization value and demonstrated the easy possibility of separating the phases with the help of a magnet. We consider that other investigations regarding the magnetic characteristics of the nanocomposite have no point in this work.
7) Linear kinetic models often underestimate the kinetic parameters, it is preferable to used non-linear kinetic models, this section 3.3.2 must be corrected. Models must be correlated using BIC parameter, please use the next reference for guideline: (1) Enhanced Removal of As(V) and Pb(II) from Drinking and Irrigating Water Effluents Using Hydrothermally Synthesized Zeolite 5A. R2 value only analyze the accuracy of each model independently, hence statistically speaking is a wrong analysis.
Answer: thanks to the reviewer for the recommendation.
Taking into account the information from the specialized literature regarding adsorption of heavy metals onto magnetic nanocomposite, linear fitting is used for kinetic studies and non-linear fitting for equilibrium studies.
As stated in the recommended article (Water 2023, 15(10), 1892; https://doi.org/10.3390/w15101892) “It is known that linear fits …are mostly used to determine the kinetic model based on the correlation factor R2, and the parameters are compared with the experimental data, which may be different due to the linearization of the equations. Therefore, to avoid inconveniences, it is recommended to perform a nonlinear fit …. “
Also, linear fitting for kinetic studies was applied in the following articles recommended by the reviewer:
(2) “Surface Adsorption Mechanism between Lead (II, IV) and Nanomaghemite Studied on Polluted Water Samples Collected from the Peruvian Rivers Mantaro and Cumbaza”
“Table 2. Linear fit parameters obtained from the pseudo-second-order kinetic model for the Cumbaza and Mantaro rivers using the RG1 sample as adsorbent. qe is the adsorbed quantity at the equilibrium time and k2 the kinetic adsorption constant”
(3) “Improved removal capacity and equilibrium time of maghemite nanoparticles growth in zeolite type 5A for Pb (II) adsorption”
“By fitting the straight line of t qt versus t, values of 1 k2q 2 e as the intercept and 1 qe as the slope were obtained”
In: Tran, H.N. (Water 2023, 15, 1231. https://doi.org/10.3390/w15061231) is mentioned: “The nonlinear optimization method is often suggested to model the adsorption kinetic model because it can minimize error function” even “the utilization of the nonlinear method can minimize some error functions during the modelling process”.
As can be seen in Table 6, the R2 values obtained for the PSO kinetic model are close to 1, and the calculated adsorption capacity is very close to the experimental data, using linear fit.
As we have seen in the specialized literature, the Bayesian information criterion (BIC) is a recently introduced statistical parameter for error analysis, and the existing papers on applying Bayesian for optimal design are related to pharmaceutical (Muller et al. 2007; Ryan et al. 2015) and biological applications (Lee and Chu, 2012; Kleinegesse and Gutmann, 2018).
In addition, according to the Montecarlo approach the application of BIC is recommended for an adequate estimation of the validity of non-linear models (Spiess and Neumeyer, 2010), in addition to the conventional assessment R2 and the chi-square test (χ2) to determine the degree of difference between the experimental data and data calculated on the model.
Eder L., Hadi D.M., Ashish G., Farooq S., Rama K., Guilherme D., Hai. T. (2021). Adsorption: Fundamental aspects and applications of adsorption for effluent treatment. DOI: 10.1016/B978-0-323-85768-0.00004-X: “For statistical evaluation of nonlinear kinetics and equilibrium models fitted, it is commonly necessary to use BIC..”
Thanks for the recommendation. In future kinetic studies, we will apply the nonlinear fitting of the experimental data and determine the statistical errors. Unfortunately, now the time allotted for the review is insufficient. Also, we saw that the BIC parameter was applied only in the first article recommended by the reviewer.
We hope that the application of the linear fit and the non-use of the BIC parameter in this study will not be an essential impediment to the publication of our results.
Thank you for understanding.
8) BIC parameter must be also employed to compare isotherm models in section 3.3.3. Table 7 must be updated with new calculations.
Answer: The BIC parameter was calculated for statistical evaluation of nonlinear equilibrium models fitted, and the data were introduced in Table 7.
In the case of the AIC and BIC parameters, the lower their value, the more reliable the model.
9) Table 8 is incomplete; the first pollutant is missing. More parent system need to be added based on magnetic nanohybrids and parent mesoporous systems: Add these references to compare: (1) Enhanced Removal of As (V) and Pb (II) from Drinking and Irrigating Water Effluents Using Hydrothermally Synthesized Zeolite 5A; (2) Surface Adsorption Mechanism between Lead (II, IV) and Nanomaghemite Studied on Polluted Water Samples Collected from the Peruvian Rivers Mantaro and Cumbaza; (3) Improved removal capacity and equilibrium time of maghemite nanoparticles growth in zeolite type 5A for Pb (II) adsorption. Table 8 must compare kinetic parameters, adsorbent dose, adsorbent size, optimum pH, equilibrium time, removal percentage, temperature, and other important adsorption parameters. A paragraph discussing and comparing must be added.
Answer: It was a typing error. We made the correction in the revised manuscript
Regarding the values included in Table 8, as mentioned in the literature, the values of maximum adsorption capacities, obtained from the analysis of the adsorption isotherms, are compared. We introduced the working conditions in Table 8.
In Table 8, the values of the maximum adsorption capacities, obtained from the analysis of the adsorption isotherms, were compared with other results reported in the literature for the use of magnetic nanocomposite as adsorbents.
We have carefully read the works suggested by the reviewer. We have included in the revised paper the works appropriate to the field investigated in the manuscript submitted by us.
10) The conclusions section must be written again it is usually one paragraph reflecting the most important findings.
Answer: The conclusion is written in order to reflect the most important findings.
Other minor points:
-Punctuation, source letter size, and grammar typos must be corrected.
Answer: Thank you for the observation. The English was checked and corrected.
-Increase the resolution of Figure 9.
Answer: Figure 9 was redone
Comments on the Quality of English Language
-Punctuation, source letter size, and grammar typos must be corrected.
Answer: Thank you for the observation. The English was checked and corrected.
Round 2
Reviewer 1 Report
Dear authors and Editor,
thank you for very detailed responses and proper account for the comments. I have enjoyed the discussion in the replies almost as much as the original text. It is very rare to see real thoughts and ideas in the reply, not just formal agree/disagree. And I have appreciated the additional tests you have conducted to verify your opinion. I wish I could be as productive as you are.
In my view, the manuscript can be accepted in the present form. I have got two minor comments, just for future consideration. I will keep the original numbering of the items not to repeat them.
8. I meant surface oxidation of carbon during the synthesis, not during adsorption of course. However, I agree with your points that too much characterization of the adsorbent sample can make the paper either too long or poorly focused on the adsorption as the major idea. I have not tracked all your earlier papers but I feel that comprehensive characterization of the produced carbon material (including more physical methods like IR spectroscopy, EDX, XPS, titration, ...) can deserve a separate report, even if it repeats some of the earlier published findings.
11. Four values are not quite many to calculate SE, I agree. But I have noticed in the papers and books that it is still quite common to show it - just to confirm graphically (in addition to the p-values) that the observed effects are significant. It is not a must but myself, I tend to add the SE limits in the plots. Up to you!
Author Response
Dear authors and Editor,
thank you for very detailed responses and proper account for the comments. I have enjoyed the discussion in the replies almost as much as the original text. It is very rare to see real thoughts and ideas in the reply, not just formal agree/disagree. And I have appreciated the additional tests you have conducted to verify your opinion. I wish I could be as productive as you are.
In my view, the manuscript can be accepted in the present form. I have got two minor comments, just for future consideration. I will keep the original numbering of the items not to repeat them.
Thanks again to the reviewer for appreciating our work and the results presented in the paper
- I meant surface oxidation of carbon during the synthesis, not during adsorption of course. However, I agree with your points that too much characterization of the adsorbent sample can make the paper either too long or poorly focused on the adsorption as the major idea. I have not tracked all your earlier papers but I feel that comprehensive characterization of the produced carbon material (including more physical methods like IR spectroscopy, EDX, XPS, titration, ...) can deserve a separate report, even if it repeats some of the earlier published findings.
Answer: We thank the reviewer for the appreciations made. We mention that our team has important contributions regarding the combustion method. We investigated the influence of several parameters on the powders characteristics such as:
- nature of the fuel: glucose, citric acid, EDTA, TWEEN80, hexamethylenetetramine
- oxidant/fuel molar ratio: fuel excess or fuel deficit
- ignition procedure of the combustion reactions: heating mantle, microwave field
- carbon and organic residues presence and chemical oxidation removal using H2O2
For this purpose, we used the most precise and modern characterization methods, including Mossbauer, XPS and FTIR spectroscopy.
The originality of the obtained results was confirmed by their utilization in numerous works published in prestigious journals indexed Web of Science-Clarivate Analytics such as:
- Lazău, R. Ianos, C. Păcurariu, D. A. Căpraru, A. Racu, V. Cornea, Combustion Synthesis of SrAl2O4: Eu2+, Dy3+, Phosphorescent Pigments for Glow-in-the-Dark Safety Markings, Nanomaterials, 13, 687, 2023 (F.I. = 5.719)
- Capraru, E. A. Moaca, C. Păcurariu, R. Ianos, R. Lazau, L. Barbu-Tudoran, Developmen and characterization of magnetic iron oxide nanoparticles using microwave for the combustion reaction ignition, as possible candidates for biomedical applications, Powder Technology, 394, 1026-1038, 2021 (F.I. = 5.64)
- G. Muntean, M.A. Nistor, R. IanoÈ™, C. Păcurariu, A. Căpraru, V.A. Surduc, , Combustion synthesis of Fe3O4/Ag/C nanocomposite and application for dyes removal from multicomponent systems, Applied Surface Science, vol. 481, pp. 825–837, 2019 ( F.I. = 7.392)
- IanoÈ™, E.A. Moacă, A. Căpraru (căs. Schulze), R. Lazău, C. Păcurariu, Maghemite, γ-Fe2O3, nanoparticles preparation via carbon-templated solution combustion synthesis, Ceramics International, 44(12), 2018, 14090-14094, 2018 (F.I.= 5.532).
- Ianos¸ C. Pacurariu, S. G. Muntean, E. Muntean, M. A. Nistor, D. Niznanský, Combustion synthesis of iron oxide/carbon nanocomposites, efficient adsorbents for anionic and cationic dyes removal from wastewaters, Journal of Alloys and Compounds 741, 1235-1246, 2018 (I.F.=6.371)
- Ianoș, R. Istratie, C. Păcurariu, R. Lazău, Solution combustion synthesis of strontium aluminate, SrAl2O4, powders: single-fuel versus fuel-mixture, Physical Chemistry Chemical Physics, Vol. 18, pp. 1150-1157, 2016 (I.F.=3.676)
- IanoÈ™, C. Păcurariu, G. Mihoc , Magnetite/carbon nanocomposites prepared by an innovative combustion synthesis technique – Excellent adsorbent materials, Ceramics International, Vol. 40(8), pp. 13649-13657, 2014 (I.F.=5.532)
- Ianoş, E.A. Tăculescu, C. Păcurariu, I. Lazău, Solution combustion synthesis and characterization of magnetite, Fe3O4, nanopowders, Journal of the American Ceramic Society, 95(7), pp.2236-2240, 2012 (I.F.=4.186)
- Four values are not quite many to calculate SE, I agree. But I have noticed in the papers and books that it is still quite common to show it - just to confirm graphically (in addition to the p-values) that the observed effects are significant. It is not a must but myself, I tend to add the SE limits in the plots. Up to you!
Answer: At reviewer suggestion we replaced Fig 5 in the revised manuscript

Reviewer 2 Report
The authors have not addressed all my comments; it seems that there is a worry about fast publication and not a care for the quality of the manuscript. Major comments are still pending.
1) Major articles, including those in prestigious journals, indicate the wrong word pattern (recent and ten-year-old recommended papers from the authors). It is the same error as with Scherrer’s formula; many authors cited the Scherrer-Debye formula (including high-impact publications). The word pattern must be changed to diffractogram according to my suggestion and previous explanation.
2) The authors mentioned in their response, "As we stated in the paper, it is difficult to distinguish between magnetite and maghemite. This can be done by Mossbauer spectroscopy or X-ray photoelectron spectroscopy. Considering the purpose of the work, it is important to demonstrate that the nanocomposite has magnetic properties regardless of whether they are due to magnetite or maghemite. That is why we consider that the results obtained based on the X-ray diffraction pattern by the PDXL2 software using the relative intensity ratio (RIR) method are sufficient."
- This is not a physically satisfying response; there are many previous works (e.g., Differentiating Nanomaghemite and Nanomagnetite and Discussing Their Importance in Arsenic and Lead Removal from Contaminated Effluents: A Critical Review) that have carefully designed and addressed the difference between nanomagnetite and nanomaghemite because adsorption is a surface mechanism, and certainly the iron state (Fe2+ or Fe3+) is extremely important and will influence the adsorption mechanism (not in detail studied here). The authors mentioned that it is only important to be magnetic; this is not correct, and they do not recognize other updated works in the area. The comment must be cited with the reference that pointed out this issue.
3) In Table 8, authors must include the equilibrium time and removal efficiency for the studied systems.
Author Response
Dear authors and Editor,
The authors have not addressed all my comments; it seems that there is a worry about fast publication and not a care for the quality of the manuscript. Major comments are still pending.
We are surprised and even offended by the statement of Reviewer 2 related to "there is a worry about fast publication and not a care for the quality of the manuscript" especially since we have a lot of experience and published works in the field (see Hirsh index of authors). We believe that the results obtained and presented in our manuscript even confirm our concern for "quality of the manuscript".
We made the corrections and submitted the review on the last day granted by the editor for sending the answers. “Please revise the manuscript according to the referees' comments and upload
the revised file within 10 days.”
1) Major articles, including those in prestigious journals, indicate the wrong word pattern (recent and ten-year-old recommended papers from the authors). It is the same error as with Scherrer’s formula; many authors cited the Scherrer-Debye formula (including high-impact publications). The word pattern must be changed to diffractogram according to my suggestion and previous explanation.
Answer: As the reviewer points out "major articles, including those in prestigious journals, indicate the wrong word pattern"... Does this mean that high-impact journals have accepted "mistakes" in published articles and we "must" to “change…according to" reviewer "suggestion"?!
We present again some works published in 2022 and 2023 in journals with an impact factor from 11,307 to 48,165 that use the name X-ray diffraction pattern:
- YoshieIshikawa, Takeshi Tsuji, Shota Sakaki, Naoto Koshizaki, Pulsed laser melting in liquid for crystalline spherical submicrometer particle fabrication– Mechanism, process control, and applications, Progress in Materials Science, Volume 131, January 2023, 101004 (F.= 48.165)
- Akun Liang, Robin Turnbull, Daniel Errandonea, A review on the advancements in the characterization of the high-pressure properties of iodates, Progress in Materials Science, Volume 136, July 2023, 101092 (F.= 48.165)
- Guowei Yang, Synthesis, properties, and applications of carbyne nanocrystals, Materials Science and Engineering: R: Reports, Volume 151, October 2022, 100692 (I.F.= 33.667)
- Pavlenko, S. Khosravi, S. ZË™ o´Å‚towska, A.B. Haruna , M. Zahid, Z. Mansurov, Z. Supiyeva, A. Galal, K.I. Ozoemena, Q. Abbas, T. Jesionowski, A comprehensive review of template-assisted porous carbons: Modern preparation methods and advanced applications, Materials Science and Engineering: R: Reports, Volume 149, June 2022, 100682 (I.F.= 33.667)
- YechuanChen, Ying Huang, Mingjie Xu, Tristan Asset, Xingxu Yan, Kateryna Artyushkova, Mounika Kodali, Eamonn Murphy, Alvin Ly, Xiaoqing Pan, Zenyuk, Plamen Atanassov, Catalysts by pyrolysis: Direct observation of transformations during re-pyrolysis of transition metal-nitrogen-carbon materials leading to state-of-the-art platinum group metal-free electrocatalyst, Materials Today, Volume 53, March 2022, Pages 58-70 (I.F.= 26.943)
- Kristy Stanley, Sean Kelly, James A. Sullivan, Effect of Ni NP morphology on catalyst performance in non-thermal plasma-assisted dry reforming of methane, Applied Catalysis B: Environmental, Volume 328, 5 July 2023, 122533 (I.F.= 24.319)
- Fabio Blaschke, Marjan Bele, Brigitte Bitschnau, Viktor Hacker, The effect of microscopic phenomena on the performance of iron-based oxygen carriers of chemical looping hydrogen production, Applied Catalysis B: Environmental Volume 327, 15 June 2023, 122434 (F.= 24.319)
- Zhe Zhao, Ye Kong, Gaoshan Huang, Chang Liu, Chunyu You, Zhijia Xiao, Hongqin Zhu, Ji Tan, Borui Xu, Jizhai Cui, Xuanyong Liu, Yongfeng Mei, Area-selective and precise assembly of metal organic framework particles by atomic layer deposition induction and its application for ultra-sensitive dopamine sensor, Nano Today, Volume 42, February 2022, 101347 (I.F.= 18.962)
- Wei Li, Zhujun Yao, Shengzhao Zhang, Xiuli Wang, Xinhui Xia, Changdong Gu, Jiangping Tu, High-performance Na3V2(PO4)2F5O0.5 cathode: Hybrid reaction mechanism study via ex-situ XRD and sodium storage properties in solid-state batteries, Chemical Engineering Journal, Volume 423, 1 November 2021, 130310 (I.F.=16.139)
- Jiang Zhao, Ning Yi, Xiaohong Ding, Shangbin Liu, Jia Zhu, Alexander, C. Castonguay, Yuyan Gao, Lauren D. Zarzar, Huanyu Cheng, In situ laser-assisted synthesis and patterning of graphene foam composites as a flexible gas sensing platform, Chemical Engineering Journal, Volume 456, 15 January 2023, 140956 (I.F.=16.139)
- Nikolaos D. Adamopoulos, Nikos G. Tsierkezos, Afroditi Ntziouni, Fu Zhang, Mauricio Terrones, Konstantinos V. Kordatos, Synthesis, characterization, and electrochemical performance of reduced graphene oxide decorated with Ag, ZnO, and AgZnO nanoparticles, Carbon, Volume 213, September 2023, 118178 (F.= 11.307)
- J. Putman, M.R. Rowles, N.A. Marks, C. de Tomas, J.W. Martin, I. Suarez-Martinez, Defining graphenic crystallites in disordered carbon: Moving beyond the platelet model, Carbon, Volume 209, 5 June 2023, 117965 (I.F.= 11.307)
Even in an article recommended by the reviewer for citation (Flores-Cano D.A., Checca-Huaman N.R.l, Castro-Merino I.L., Pinotti Ca.N., Passamani E.C., Litterst F.J., Ramos-Guivar J.A., Progress toward Room-Temperature Synthesis and Functionalization of Iron-Oxide Nanoparticles, Int. J. Mol. Sci. 2022, 23(15), 8279), the name XRD patterns is used: “Figure 1. Rietveld refined XRD patterns of the M1–M6 samples. Black circles are the experimental data, red lines indicate calculated XRD pattern, and olive and purple vertical lines represent the Bragg positions of the γ-Fe2O3 and α-FeOOH phases, respectively”
We believe that the "obligation" to modify the diffractogram just because the reviewer wants it, given that the name X-ray diffraction pattern is used in top scientific journals, is unethical and has no scientific justification.
2) The authors mentioned in their response, "As we stated in the paper, it is difficult to distinguish between magnetite and maghemite. This can be done by Mossbauer spectroscopy or X-ray photoelectron spectroscopy. Considering the purpose of the work, it is important to demonstrate that the nanocomposite has magnetic properties regardless of whether they are due to magnetite or maghemite. That is why we consider that the results obtained based on the X-ray diffraction pattern by the PDXL2 software using the relative intensity ratio (RIR) method are sufficient."
- This is not a physically satisfying response; there are many previous works (e.g., Differentiating Nanomaghemite and Nanomagnetite and Discussing Their Importance in Arsenic and Lead Removal from Contaminated Effluents: A Critical Review) that have carefully designed and addressed the difference between nanomagnetite and nanomaghemite because adsorption is a surface mechanism, and certainly the iron state (Fe2+ or Fe3+) is extremely important and will influence the adsorption mechanism (not in detail studied here). The authors mentioned that it is only important to be magnetic; this is not correct, and they do not recognize other updated works in the area. The comment must be cited with the reference that pointed out this issue.
Answer: Since it is not ethical for a reviewer to impose the citation of his own works, we have cited in the revised manuscript other works in which XPS techniques and Mossbauer spectroscopy were used to differentiate between magnetite and maghemite, namely:
- Joos, Rümenapp C., Wagner F.E., Gleich B., Characterization of iron oxide nanoparticles by Mössbauer spectroscopy at ambient temperature, Journal of Magnetism and Magnetic Materials, 2016, 399, 123-129.
- Ianoş R., Tăculescu E.A., Păcurariu C., Lazău I., Solution combustion synthesis and characterization of magnetite, Fe3O4, nanopowders, Journal of the American Ceramic Society, 2012, 95(7), 2236-2240.
It is up to the Editor to consider whether it is necessary and ethically correct to introduce articles recommended by reviewer 2 since the recommendations for reviewers are: “Reviewers should avoid recommending more than 2 citations of their work”, “Reviewers have a responsibility to promote ethical peer review”, "The anonymity of reviewers is strictly preserved from the authors, unless a reviewer voluntarily signs their comments to authors."
3) In Table 8, authors must include the equilibrium time and removal efficiency for the studied systems.
Answer: even if we did not find such a thing in the specialized literature in articles related to "adsorption of metal ions", at the reviewer's recommendation, we entered the "required" values in Table 8 in the revised form of the manuscript.
